# Nanoformulated Terpenoids in Cancer: A Review of Therapeutic Applications, Mechanisms, and Challenges

**DOI:** 10.3390/cancers17183013

**Published:** 2025-09-16

**Authors:** Arunagiri Sharmila, Priyanka Bhadra, Chandra Kishore, Chinnadurai Immanuel Selvaraj, Joachim Kavalakatt, Anupam Bishayee

**Affiliations:** 1Department of Biotechnology, School of Biosciences and Technology, Vellore Institute of Technology, Vellore 632 014, Tamil Nadu, India; 2Department of Biological Sciences, Bose Institute, Kolkata 700 091, West Bengal, India; 3Division of Pulmonary, Critical Care and Sleep Medicine, Department of Medicine, Icahn School of Medicine at Mount Sinai, New York, NY 10029, USA; 4Tisch Cancer Institute, Icahn School of Medicine at Mount Sinai, New York, NY 10029, USA; 5Department of Genetics and Plant Breeding, VIT School of Agricultural Innovations and Advanced Learning, Vellore Institute of Technology, Vellore 632 014, Tamil Nadu, India; 6Department of Pharmacology, College of Osteopathic Medicine, Lake Erie College of Osteopathic Medicine, Bradenton, FL 34211, USA; jkavalakat75461@med.lecom.edu

**Keywords:** terpenoids, nanoformulations, drug delivery, anticancer effects, molecular mechanisms

## Abstract

Cancer remains a significant global health concern for which safe and effective treatment is needed. Terpenoids are natural substances derived from plants that possess the ability to prevent cancer by inhibiting cell growth, inducing cell death, and preventing metastasis. Nevertheless, the clinical applicability of terpenoids is restricted by their poor bioavailability and low water solubility. This review addresses the ways in which nanoformulation techniques can improve terpenoid distribution and efficacy. It discusses several nanocarriers that facilitate the regulated release of terpenoids as well as enhance their solubility, stability, and targeted delivery. The review also highlights current issues in clinical translation as well as emerging developments, such as the integration of artificial intelligence in nanomedicine. In summary, nanoformulated terpenoids represent a potential approach for cancer treatment by combining bioactive natural compounds with advanced drug delivery systems.

## 1. Introduction

Cancer remains a critical health burden across the world, affecting global populations and healthcare systems with high incidence and fatality rates. The estimated 20 million new cases of cancer in 2022 are anticipated to rise by 77% to over 35 million cases in 2050 worldwide [1]. It remains the second-leading cause of mortality globally, accounting for around 8.97 million deaths per year, and is expected to increase to almost 18.63 million by 2060. Men are marginally more likely than women to develop cancer, with a lifetime risk of 20.2% up until the age of 74. Overall, stomach, liver, and lung cancers have the largest mortality rates, with breast cancer being more common in women and lung cancer in males [2]. It is the foremost cause of mortality among individuals under 85 in the United States. In 2025, the US is projected to report approximately 2,041,910 new cancer cases and 618,120 cancer fatalities [3]. India, the most populous country in the world, faces an increasing cancer prevalence with anticipated rises in both incidence and death rates. Cancer incidence is estimated to increase from 529.40 cases per 100,000 population in 2022 to 549.17 cases by 2031, while fatalities are anticipated to rise from 71.52 to 73.00 per 100,000 during the same period. Numerous causes, including fast urbanization, an aging population, more sedentary lifestyles, consumption of unhealthy food choices, and increase in air pollution, are some of the causes for this increase in cancer incidence in India [4]. These statistics underscore the urgent need for effective strategies in cancer detection, prevention, and treatment to address these escalating global health challenges.

Conventional cancer therapies include surgery, chemotherapy, radiation, targeted therapy, immunotherapy, and hormone therapy. Despite their effectiveness, radiation and chemotherapy have a significant risk of recurrence, severe side effects, drug-specific toxicities, and may lead to the progression of secondary cancers. Targeted therapies have improved precision but remain limited by multidrug resistance and dermatologic adverse effects. Immunotherapy has shown great promise in reducing recurrence and metastasis. It is less effective against solid tumors and entails the risk of autoimmune diseases. Solid tumors frequently produce a thick extracellular matrix, which hinders the ability of immune cells to infiltrate [5,6]. Surgery is frequently used when cancer is in its early stages and is usually not recommended in advanced stages [7]. These limitations underscore the urgent need for safer and more effective therapeutic strategies. It is essential to develop novel treatments that maintain efficacy comparable to other treatments while minimizing adverse effects and offering improved safety and tolerability profiles. The phytochemicals and plant metabolites offer potential strategies to increase therapeutic effectiveness and reduce adverse effects. Several reports have exhibited the anticancer efficacy of phytocompounds, such as phenols [8], alkaloids [9], glycosides [10], terpenoids [11], saponins [12], and sulfur-containing compounds [13]. Paclitaxel [14], vincristine [15], camptothecin [16], and etoposide [17] are some of the successfully plant-derived drugs that are popular anticancer agents in the treatment against various types of cancer, drawing attention in the research community toward further potential plant-based drug alternatives and therapies.

Plant-derived anticancer compounds can be a better alternative to conventional approaches, and terpenoids are promising choices. Terpenoids, derived from oxygenated terpenes, are crucial bioactives promoting the global use of natural remedies. The mevalonate pathway in plants plays a major role in terpenoid synthesis, while certain other terpenoids can also be derived from aquatic organisms [18]. They promote holistic, sustainable physiological benefits by providing antibacterial, anti-inflammatory, and therapeutic properties [19]. There are currently about 90,000 known terpenoids, with the majority of them obtained from plants, microbes, and marine sources [20]. The diverse range of intriguing terpenoid structures synthesized by marine species can be identified by their different structural characteristics. The metabolic products of aquatic organisms have a wide range of well-defined structural groups, including chamigrene, cembrane, and amphilectane, as well as unique functional categories, including isothiocyanate, isonitrile, isocyanate, dichloroimine, and halogen-containing groups [21].

Terpenoids are essential in a variety of sectors, including medicine [22], agriculture [23], cosmetics [24], food processing [25], and biofuels [26], owing to their diverse range of special physicochemical activities and biological roles. Terpenoids possess a significant role in different pharmacological applications (Figure 1). They exhibit antimicrobial effects on bacteria, facilitated by the terpenoids’ ability to induce cell lysis and prevent DNA and protein synthesis [27,28]. Terpenoids demonstrated excellent antioxidant activity, with the multiple mechanisms of action not limited to electron transfer, hydrogen transfer, and singlet oxygen quenching [29,30,31]. They have demonstrated significant anti-inflammatory [32,33,34], antidiabetic [35], antinociceptive [36], and anticancer properties [11,37,38]. Despite their promise of therapeutic qualities, terpenoids exhibit limited clinical applicability due to their non-specific distribution, low intestinal permeability, quick elimination, and poor water solubility. These restrictions make it more difficult to directly convert preclinical findings into clinical outcomes. Various advancements, such as nanocarrier-based drug delivery systems, present a promising solution and increase their total therapeutic potential by improving solubility, stability, pharmacokinetics, and biodistribution, while reducing toxicity [39]. In comparison with other plant phytochemicals, terpenoids further present distinct advantages as candidates for anticancer nanoformulations. Their structural diversity confers broad-spectrum anticancer activity, while their generally low toxicity profile enhances safety [40]. Moreover, the inherent lipophilicity of terpenoids facilitates efficient encapsulation within nanocarriers and promotes cellular uptake. Importantly, terpenoids have demonstrated potential in overcoming drug resistance and can synergize with other therapeutic agents, positioning them as highly promising core components for nanocarrier-based cancer therapies [41].

Nanomedicine is essential in the development of precise treatments and drug formulations, as well as in regulating the release of drug delivery systems. Drug nanocarriers offer a promising strategy in cancer therapy by improving circulation and therapeutic index while reducing toxicity to normal cells [42]. Nanotechnological approaches, especially nanoformulations, are an advanced method for enhancing the solubility and stability of phytoconstituents, enhancing absorption, safeguarding them from the body’s early deterioration, and extending their half-life. Additionally, these nanoparticles exhibit a higher penetration and absorption effect in disease tissues, hinder early contact with the biological environment, improve cellular uptake, and reduce toxicity by demonstrating a significant differential uptake capability in targeted cells [43]. The clinical use of phytocompounds is limited by several physicochemical and pharmacokinetic restrictions, including low water solubility, bioavailability, and poor selectivity. To improve their anticancer efficacy, numerous nanodrug delivery systems are being developed [44,45]. Nano-based delivery systems in cancer have the benefit of improved permeability and retention, lowering tumor hypoxia and resistance, and focusing on tumor-associated factors, such as fibroblasts, endothelial cells, and immune cells [46].

The therapeutic potential of terpenoids is limited by poor solubility, rapid metabolism, low bioavailability, and toxicity to normal cells, which hinder their effectiveness in vivo and restrict their medicinal use. Nanoformulations offer a promising solution by protecting terpenoids from degradation, enhancing pharmacokinetic properties, and improving oral absorption [47]. Through nanotechnology-based delivery systems, terpenoids achieve better systemic availability, targeted delivery, and enhanced therapeutic efficiency across various cancers. They enable targeted delivery to specific tissues, thereby reducing adverse effects. In addition, nanoencapsulation allows the co-delivery of multiple bioactive molecules to achieve synergistic anticancer effects [48]. Furthermore, terpenoid formulations based on nanotechnology have demonstrated encouraging outcomes in cancer therapeutics, with fewer side effects and enhanced effectiveness [49]. Nanotechnology provides innovative strategies to overcome the major limitations of terpenoids, including poor solubility, low bioavailability, and limited cellular uptake, by enhancing their target specificity and therapeutic efficacy. Although the synthesis and classification of terpenoids in anticancer research have been extensively studied, the application of advanced nanotechnology-based delivery systems to improve their dispersion, stability, and biological uptake remains relatively underexplored.

The aim of this review is to provide a comprehensive and critical analysis of terpenoid-loaded nanoformulations specifically developed for anticancer applications. The novelty of this review lies in presenting an integrative perspective on recent advancements and therapeutic significance in anticancer treatment. In contrast to other reviews that have primarily addressed terpenoids or nanocarriers separately [11,46], this work brings together both aspects to emphasize their combined potential in oncology, with particular focus on enhancing targeted drug delivery to tumor sites. Nanoformulated terpenoids enable selective tumor accumulation, reduce systemic toxicity, and maximize clinical efficacy, thereby underscoring their translational promise in cancer therapy. In addition, this review addresses translational challenges and evaluates diverse nanocarrier platforms. Together, these perspectives provide timely insights into the development of terpenoid-based anticancer nanomedicines by integrating natural product pharmacology, nanotechnology, and cancer therapeutics.

## 2. Classification and Biosynthesis of Terpenoids

Terpenes are simple hydrocarbons that include limonene, terpinene, pinene, p-cymene, and myrcene, while terpenoids are modified forms with oxygenated hydrocarbons or oxidized methyl groups [40]. They are classified by C5 units into hemiterpenes, monoterpenes, sesquiterpenes, diterpenes, sesterterpenes, triterpenes, tetraterpenes, and polyterpenes [50]. Hemiterpenes (C_5_H_8_) act as plant defense signals, while monoterpenes (C_10_H_16_) and sesquiterpenes (C_15_H_24_) show ecological roles and pharmacological effects, including antiviral, antibacterial, anti-inflammatory, and anticancer activities [51,52]. Diterpenes (C_20_H_32_), sesterterpenes (C_25_H_40_), and triterpenes (C_30_H_48_) are widely distributed in nature and exhibit antioxidant, antimicrobial, and anticancer properties [53,54]. Tetraterpenes (C_40_H_64_) such as carotenoids have immunomodulatory and cardioprotective effects [55], while polyterpenes (C_40_H_64_)_n_, including natural rubber, occur as plant latex [52]. Their classification by isoprene units, carbon number, and biological activity is presented in Table 1 and Figure 2.

Terpenoids are synthesized in vivo via two distinct pathways, namely the mevalonate pathway (MVA) and the 2-C-methyl-D-erythritol-4-phosphate (MEP)/1-deoxy-D-xylulose-5-phosphate pathway (DXP) (Figure 3) [56]. Both generate the C5 precursors IPP and DMAPP, which undergo rearrangements and cyclizations to form terpenoid backbones. The MVA pathway, used by eukaryotes, archaea, and various bacteria, provides precursors for triterpenes, sesquiterpenes, and phytosterols, while the MEP pathway in cyanobacteria, algae, and plant chloroplasts yields carotenoids, monoterpenes, and diterpenes [57,58]. Terpenoid biosynthesis involves three stages: (i) isopentenyl diphosphate (IPP)/dimethylallyl diphosphate (DMAPP) formation; (ii) precursor synthesis of geranyl diphosphate (GPP), farnesyl diphosphate (FPP), and geranylgeranyl diphosphate (GGPP); and (iii) terpene skeleton formation and modification [59]. GPP, FPP, and GGPP act as key intermediates for mono-, sesqui-, di-, and tetraterpenes, as well as sterols and carotenoids [60]. Further structural diversity arises from modifications by terpene synthases and enzymes such as cytochrome P450, leading to a wide range of chemical properties and biological activities [61].

**Table 1 cancers-17-03013-t001:** Terpenoid classification according to carbon atoms, isoprene unit, compound name, and biological activities.

Class	No. of C Atoms	No. of Isoprene Units (C_5_H_8_)_n_	Name of Compounds	Biological Activities	Ref.
Hemiterpenes	C_5_	1	Isovaleric acid, senecioic acid, angelic acid, and tiglic acid	Constituents in medicines, rubber, potential biofuels, and flavors	[62]
Monoterpenes	C_10_	2	Geraniol, pinene, linalool, sabinene, borneol, menthol, myrcene, and limonene	Constituents in cosmetics, food, and pharmaceuticals. It shows antimicrobial, anti-inflammatory, antitumor, and cardioprotective properties	[63]
Sesquiterpenes	C_15_	3	Germacrane, eudesmane, β-cadinene, β-caryophyllene, artemisinin, farnesol, geosmin, and cadinene	Possess antiviral, antibacterial, antidiabetic, antiobesity, and anti-inflammatory properties	[55,64]
Diterpenes	C_20_	4	Andrographolide, sclareol, excavatolide B, kirenol, carnosic acid, phytol, retinoic acid, and oridonin	Possess antioxidant, anti-inflammatory, immune-modulatory, and antirheumatoid arthritis action	[53]
Sesterterpenes	C_25_	5	Ophiobolin A, bilosespene A, merochlorin A, secoemestrin D, sterigmatocystin, leucosceptroid A, and colquhounoid A	Exhibits anticancer, antifeedant, and antifungal activities	[51,65]
Triterpenes	C_30_	6	Squalene, lupeol, lanostane, oleanolic acid, betulinic acid, ursolic acid, dammarane, and ginsenoside	Possess antibacterial, antifungal, anti-inflammatory, antioxidant, anticancer, antiviral, and chemopreventive properties	[66]
Tetraterpenes	C_40_	8	Carotenoid, lutein, astaxanthin, fucoxanthin, lycopene, phytoene, and β-carotene	Immunomodulatory, cardioprotective, and anticancer activities	[67]
Polyterpenes	>C_40_	>8	Natural rubber and resins	Used as sealants, hot-melt adhesives, and pressure-sensitive adhesives, chewing gum contains certain polyterpene resins	[68]

## 3. Mechanistic Insights into the Anticancer Properties of Terpenoids

Interest in terpenoids has grown because of their unique structural characteristics, strong anticancer activity, and potential to serve as key molecules in the advancement of effective antitumor medications [61]. Terpenoids display anticancer properties by targeting various molecular mechanisms involved in oncogenesis and cancer progression and by inhibiting cell growth mechanisms. They suppress the initial phases of carcinogenesis through cell cycle arrest induction, inhibition of cancer cell differentiation, and initiation of cancer cell death. By targeting various intracellular signaling mechanisms, some terpenoids can inhibit angiogenesis and metastasis in the advanced phases of cancer growth [11,69]. Terpenoids possess significant anticancer properties because they target multiple phases of cancer growth, including cellular invasion, cell death, autophagy, and proliferation (Figure 4). Terpenoids induce cell cycle arrest and apoptosis by activating p53/p21, inhibiting cyclin B/cdk1, and triggering mitochondrial pathway-related molecules, such as Bax/Bak, cytochrome c (cyt. c), caspase-9, caspase-3, and caspase-7. They also downregulate Bcl-2/Bcl-xL and suppress Janus kinase 2/signal transducer and activator of transcription 3 (JAK2/STAT3), as well as nuclear factor-κβ (NF-κβ) signaling, reducing matrix metalloproteinase-9 (MMP-9) and cell invasion (Figure 4A). Terpenoids promote autophagy by activating AMP-activated protein kinase (AMPK), reactive oxygen species (ROS), and mitogen-activated protein kinase (MAPK) pathways, as well as upregulating Beclin 1, autophagy-related protein 5/7 (ATG5/7), and microtubule-associated protein 1A/1B-light chain 3 (LC3-II), while inhibiting phosphatidylinositol-3-kinase (PI3K)/protein kinase B (Akt)/mammalian target of rapamycin (mTOR), mesenchymal–epithelial transition factor (c-Met), and glucose transporter 1 (Glut-1) to reduce proliferation and glucose uptake. By suppressing heat shock factor 1 (HSF-1), heat shock protein 70 (HSP70), and NF-κβ, they inhibit tumor growth and show therapeutic potential against drug-resistant cancers (Figure 4B).

Terpenoids can prevent the erroneous processes implicated in cancer by regulating cell cycle arrest, apoptosis, and autophagy rather than activating through individual pathways in a coordinated and synergistic manner. These processes are connected to growth inhibition, cellular stress responses, and programmed cell death through similar signaling mediators such as p53, Bcl-2 family proteins, caspases, MAPK, and NF-κβ. Terpenoids eradicate cancerous cells by inducing mitochondrial and receptor-mediated apoptosis, restrict unchecked proliferation by enforcing cell cycle checkpoints, and improve cellular quality control and apoptosis sensitization in cancer cells by modifying autophagy. These pathways also interact to provide a cumulative inhibitory impact that increases cytotoxicity and gets past resistance mechanisms. Research demonstrates that terpenoids can increase tumor suppression by simultaneously inducing apoptosis, controlling autophagic flux, and stopping cell cycle progression in the same biological environment. Furthermore, their combination with conventional chemotherapeutics potentiates these synergistic actions, offering a promising strategy to improve overall treatment efficacy [11,70,71].

## 4. Challenges with Natural Terpenoids and Advantage of Nanoformulations

Terpenoids have shown significant anticancer activities via oxidative stress reduction, cell growth inhibition, apoptosis induction, angiogenesis suppression, metastasis suppression, and inflammation resolution. However, their clinical application is mostly hindered by limitations, including low aqueous solubility, low systemic availability, and quick elimination from the body. Recently, phyto-nanomedicine has received significant focus, which deals with phytochemical-loaded nanoformulations for human healthcare. Evidence indicates that by utilizing the benefits of drug delivery via nanoparticles, terpenoids can overcome their limitations and become viable anticancer treatment candidates through nanoformulation. This approach enhances their solubility and stability, extends systemic circulation, and enhances site-specific delivery to tumor sites. For instance, celastrol, a pentacyclic terpenoid with anticancer activity, suffers from poor aqueous solubility and low bioavailability. Incorporating celastrol into nanocarriers has been shown to increase its therapeutic efficacy by improving its delivery and accumulation in tumors [72,73,74]. Terpenoids’ hydrophobic characteristic restricts the drug’s therapeutic application, and increasing the production of hydrophilic terpenoids to supplement terpenoid solubility may improve their pharmacokinetic characteristics, including plasma content in the blood and oral bioavailability. Furthermore, structural changes, such as less epoxidation and allylic methyl group oxidation, might lessen the toxicity of terpenoids. As a result, solid lipid nanoparticles incorporated with terpenoids have improved stability and bioavailability while retaining their anticancer properties [11]. Moreover, nanoformulations can facilitate the co-delivery of terpenoids with other chemotherapeutic agents, with the potential to enhance synergy and combat multidrug resistance in cancer. This approach could improve the overall efficacy of cancer therapy [44].

Nanocarrier systems offer specific solutions to overcome the limitations of terpenoids, including poor solubility, instability, and low bioavailability. For instance, solid lipid nanoparticles loaded with thymoquinone improved oral bioavailability [75], while liposomes loaded with ursolic acid increased solubility and sustained release [76]. Betulinic acid encapsulated in poly(lactic-co-glycolic acid) (PLGA) nanoparticles achieved sustained release and greater tumor accumulation [77], and celastrol in polymeric micelles showed markedly improved solubility and therapeutic efficacy [78]. These examples demonstrate that nanoformulations not only resolve the inherent drawbacks of terpenoids but also maximize their anticancer potential through improved delivery and therapeutic outcomes. Numerous research organizations have shown interest in nanoformulations as possible drug delivery vehicles. Nanoformulations involve incorporating these terpenoids into nanoparticles, which can overcome the aforementioned limitations in bioavailability and solubility and enhance therapeutic potential. Nanoformulations offer key benefits in cancer therapeutics by enhancing the delivery of terpenoids. They improve water solubility and protect compounds from degradation, increasing stability in the body. Nanoparticles also enhance bioavailability by promoting better absorption and longer circulation time. Through surface modifications, they enable targeted distribution to cancer cells, reducing side effects on healthy tissue. Additionally, they provide controlled release, maintaining therapeutic levels over time. These features increase the efficacy and safety of terpenoids, enabling them to have more potential for cancer therapy [44,47,79].

The high surface-to-mass ratio of nanoparticles (between 10 and 100 nm) facilitates their easy absorption and passage through the cellular plasma membrane and permits them to attach, absorb, and deliver drugs, RNA, proteins, DNA, and identifying compounds into the cell [80]. Nanoparticles offer the benefits of target-specific drug administration, facilitating the route of water-insoluble drugs across cell membranes, and enabling the simultaneous administration of one or more therapeutic or diagnostic agents. Earlier studies have shown that terpenoids incorporated into nanoformulations have improved pharmacokinetics, biodistribution, and solubility, all of which contribute to their higher therapeutic index [81,82]. Nanoformulations promote efficient uptake by cancer cells due to their increased stability, targeted delivery, and therapeutic activity, and induce ROS, mitochondrial damage, and caspase-mediated apoptosis. The various advantages of nanoformulated terpenoids-derived anticancer medications are illustrated in Figure 5.

## 5. Various Processes of Preparing Nanoformulations Containing Terpenoids

Nanocarriers, such as liposomes, micelles, dendrimers, solid lipid nanoparticles, inorganic carriers, and carbon nanotubes, facilitate targeted delivery of bioactive compounds to tumors, mainly through the enhanced permeability and retention effect caused by poor lymphatic drainage [83,84,85]. Depending on synthesis methods, these can be structured as nanospheres or nanocapsules to control drug encapsulation and release, thereby improving drug accumulation at tumor sites, while reducing systemic toxicity. For terpenoids, nanoformulation plays a crucial role in overcoming hydrophobicity by enhancing solubility, stability, and bioavailability. A variety of nanocarrier approaches have been developed for terpenoid delivery, each offering unique benefits and limitations. The different types of nanocarriers for anticancer treatment are illustrated in Figure 6.

Several other nanocarrier systems have been employed to improve the delivery of terpenoids, each with distinct preparation processes and therapeutic implications. Self-emulsifying drug delivery systems (SEDDS) generate fine oil-in-water emulsions to enhance solubility and oral absorption. Polymeric nanoparticles are commonly prepared by emulsion–evaporation methods, allowing sustained and targeted release. Pickering emulsions use solid particles, such as silica nanoparticles, to stabilize emulsions without surfactants, forming core–shell structures. Liposomes, composed of phospholipid bilayers, encapsulate both hydrophilic and lipophilic terpenoids to improve systemic availability. Solid lipid nanoparticles (SLNs) and nanostructured lipid carriers (NLCs) utilize solid and mixed lipid matrices to provide controlled release and higher stability. Other advanced carriers, such as ethosomes, phytosomes, niosomes, and invasomes, employ phospholipid–ethanol or surfactant-based vesicles to enhance skin permeability, oral absorption, and transdermal delivery [86,87,88].

Nanoformulation of terpenoids through various nanodelivery systems significantly improves their solubility, stability, bioavailability, and targeted delivery. Liposomes and polymeric nanoparticles may serve as promising carriers for solid tumors, such as breast, lung, and ovarian cancers, due to their ability to enhance solubility and sustain release. SLNs and NLCs appear more suitable for highly lipophilic terpenoids requiring prolonged circulation, while SEDDS and Pickering emulsions show potential in gastrointestinal cancers. Ethosomes and phytosomes can be considered preferable for skin- and liver-targeted therapies, whereas inorganic nanoparticles may hold greater value for theranostic applications. These approaches may enhance the therapeutic efficacy of terpenoids in cancer therapeutics, offering controlled release and reduced toxicity. The comparative overview of nanocarriers and their route of administration, advantages, and disadvantages is presented in Table 2.

## 6. Mechanistic Insights and Anticancer Applications of Nanoformulated Terpenoids

Nanoformulated phytocompounds exert enhanced anticancer effects through multiple mechanisms driven by improved delivery, stability, and bioavailability. Terpenoid-loaded nanoformulations can stimulate natural immune system components (cytokines and antibodies), which work together to combat and defeat cancer at the molecular level [92]. This strategy increases the survival rate of cancer patients by offering appropriate chances for multimodal, site-specific drug delivery to the tumor locations. Terpenoids are encapsulated within nanoparticles whose surfaces are functionalized with cell-specific ligands that recognize and attach to receptors on cancer cells. This targeting facilitates internalization of nanoparticles by endocytosis, followed by endosomal escape, resulting in the controlled release of terpenoids inside the cancer cells [93]. Nanoformulations can improve terpenoid safety by lowering the toxicity and general intake of the compound in the body through the direct delivery of terpenoids to particular tissues or cells. This method of targeted distribution enhances terpenoids’ safety profile and reduces adverse effects [94].

The anticancer effects of nanoformulated terpenoids encompass multiple mechanisms, including inhibition of tumor cell growth, induction of apoptosis, DNA damage, and mitochondrial disruption. When encapsulated in nanosized carriers, such as liposomes, polymeric nanoparticles, or solid lipid nanoparticles, their therapeutic efficacy is further enhanced. Additionally, co-delivery systems combining terpenoids with conventional chemotherapeutics provide synergistic antitumor effects with minimized side effects [95]. These nanoformulations trigger cancer cell apoptosis through mitochondrial dysfunction and caspase activation, induce sustained cell cycle arrest, and modulate autophagy, thereby disrupting key survival processes. It promotes enhanced tumor targeting and accumulation via the enhanced permeability and retention effect (EPR) which reduces systemic toxicity while overcoming multidrug resistance by bypassing efflux pumps and regulating critical signaling pathways [96]. Overall, nanoformulated terpenoids represent a potent strategy to improve therapeutic specificity and efficacy across various types of cancer. The schematic representation of the anticancer mechanism of nanoformulated terpenoids is shown in Figure 7.

The anticancer potential of terpenoids, including their antiproliferative, proapoptotic, and immunomodulatory activities, is often limited by poor solubility and low systemic availability. Nanoformulations overcome these limitations by enabling targeted delivery with reduced side effects, making terpenoids promising therapeutic agents for various cancers. In the following sections, we discuss different anticancer properties and their mechanisms of action of various nanoformulated terpenoids. These studies are also summarized in Table 3 and Table 4 based on in vitro and in vivo results, respectively.

### 6.1. Monoterpenoids

Numerous monoterpenes are cytotoxic to a variety of tumor cell lines. Nevertheless, because of their lipophilic nature and requirement for high concentrations to produce antitumor effects, their application in human cancer research is restricted. Consequently, research has focused on enhancing the anticancer efficacy of monoterpenes compared to their free form by using drug delivery systems [97]. Poly(methyl methacrylate) nanoparticles containing α-terpineol (5 µg/mL) exhibited notable cytotoxic activity against human (SK-MEL-28) and mouse (B16-F10) melanoma cell lines, with inhibitory percentages of 43.8 ± 1.8% and 40.8 ± 7.7%, respectively, indicating the potential of these nanoparticles for melanoma treatment [98]. The increased potency can be attributed to improved cellular uptake and sustained release of α-terpineol from the nanoparticles. The carvacrol nanoemulsions (CEN) exhibited an increase in ROS production against A549 cells at 100 µg/mL, which in turn induced cytochrome c (cyt. c) release, caspase activation, and the regulation of critical factors, including p-JNK, Bax, and Bcl-2. This ultimately led to intrinsic apoptosis mediated by mitochondria [99]. CEN decreased MMP levels and suppressed tumor growth while inhibiting angiogenesis by downregulating cyclooxygenase-2 (COX-2), VEGF, CD31, and MAPK pathway proteins in both in vitro and in vivo models [100], highlighting how nanoencapsulation potentiates apoptotic and anti-angiogenic signaling. NLCs loaded with citral, a monoterpenoid, showed higher in vitro anti-metastatic efficacy than free citral against MDA-MB-231 breast cancer cells, and in vivo, they prevented metastasis in TNBC 4T1-challenged mice [101]; the enhanced effect is likely due to improved stability, cellular uptake, and controlled release.

Lipid-based nanocarriers loaded with perillyl alcohol demonstrated notable cytotoxicity against brain cancer cell lines, indicating that lipid encapsulation facilitates brain delivery across the blood–brain barrier [102]. The anticancer agent geraniol was conjugated with hyaluronic acid via a disulfide bond to produce a targeted delivery system, which significantly increased cytotoxicity in PC-3 cells and inhibited tumor growth in male nude mice, suggesting receptor-mediated targeting enhances efficacy [103]. Linalool nanoparticles (size = 93 nm), prepared via a self-nanoemulsifying drug delivery system and administered at 100 mg/kg in BALB/c mice, increased caspase-3 activity, ROS production, apoptosis, and decreased MMP potential and tumor growth in epithelial ovarian carcinoma [104]. Menthol nanoliposomes enhanced lymph node homing of dendritic cell-based anti-tumor vaccinations, decreasing tight junction protein expression and increasing CCR7-mediated CCL21 secretion, thereby improving immune cell targeting [105]. These studies collectively indicate that monoterpenoid-based nanoformulations enhance cytotoxicity, apoptosis, tumor suppression, antimetastatic activity, and immune targeting primarily by improving cellular uptake, stability, and controlled release, while also modulating key molecular pathways.

### 6.2. Sesquiterpenoids

Sesquiterpenes are potent compounds in the treatment of cancer. Despite these advantages, they have demonstrated several pharmacological limitations due to poor physicochemical properties and unfavorable pharmacokinetics. These challenges can be mitigated through nanoformulation strategies [106]. Nerolidol-loaded solid lipid nanoparticles (NR-LNPs) demonstrated half-maximal inhibitory concentration (IC_50_) values of 38.5 μg/mL against the Caco-2 cell line following 24 h incubation. By triggering apoptosis, NR-LNPs limited cancer cell growth to 4.2-fold more effectively than free nerolidol, highlighting how nanoparticle encapsulation improves cellular uptake and bioavailability [107]. Similarly, zerumbone-loaded nano-lipid carriers exhibited potent anticancer activity against acute human lymphoblastic leukemia cells through activation of the mitochondrial apoptotic pathway, suggesting enhanced drug delivery and mechanistic targeting [108]. Costunolide-loaded bilosome nanoparticles demonstrated an IC_50_ of 6.20 µM versus 15.78 µM for free costunolide, and effectively downregulated BCL2 while upregulating CASP3, TP53, and BAX, showing that nanocarriers can amplify apoptotic signaling [109]. PLGA-PEG-helenalin nanoencapsulation (size = 35–85 nm) reduced IC_50_ from 1.3 μM (free drug) to 0.14 μM against breast cancer cell lines, demonstrating the role of controlled release and improved stability [110].

Co-loaded niosomal nanoparticles with curcumin and artemisinin (size = 210.10 ± 13.04 nm) markedly enhanced colorectal cancer inhibition by modulating BCL2, RB, and CYCLIN D1, and upregulating BAX, FAS, and p53 [111]. Furthermore, a niosomal drug transport system loaded with farnesol and gingerol (size = 184.4–338.4 nm) showed a notable inhibitory effect against MCF-7 and SKBR3 cell lines through apoptosis induction and cell cycle regulation, emphasizing targeted intracellular delivery by upregulating BAX, CASP3, CASP9, and p21, and downregulating BCL2, CCNE, CDK4, CCND1, and HER2 [112]. Copper oxide (CuO)/titanium dioxide (TiO_2_)-chitosan-farnesol nanocomposites generated ROS, decreased MMP, and initiated apoptosis in SK-MEL-3 cells [113]. Similarly, cisplatin’s anticancer action against hepatocellular carcinoma (HepG2) cells was enhanced by the co-encapsulated PLGA nanoform of farnesol and cisplatin (size = 115 ± 2.5, ζ potential = −17 ± 1.5 mV) at 1.5 µg/mL [114]. In another study, human serum albumin was utilized to encapsulate artemether (a chemical derivative of artemisinin), resulting in albumin nanoparticles (size = 171.3 ± 5.88 nm) with a ζ potential of −19.1 ± 0.82 mV. It demonstrated decreased colon cancer growth in BALB/c mice by lowering interlukin-4 (IL-4) production and enhancing interferon γ (IFN-γ) cytokine release [115]. Collectively, these findings indicate that sesquiterpenoid-based nanoformulations enhance cytotoxicity, apoptosis, ROS generation, and immune modulation, providing more effective and targeted anticancer therapy through improved delivery, stability, and mechanistic engagement.

### 6.3. Diterpenoids

Advanced drug delivery systems play a crucial role in improving the therapeutic efficacy of diterpenes against cancer and overcoming challenges, such as selective toxicity, low bioavailability, and rapid metabolism [116]. Membrane protein-chimeric liposomes loaded with triptolide (size = 157.3 ± 1.31 nm) demonstrated IC_50_ values of 17.25 and 33.97 ng/mL against Hep3B and Huh7 cells, respectively, illustrating enhanced cellular uptake and antitumor efficacy in hepatocellular carcinoma [117]. Celastrol-loaded poly(ethylene glycol)-block-poly(ɛ-caprolactone) nanopolymeric micelles suppressed SO-Rb 50 cell proliferation in a concentration-dependent manner (IC_50_ = 17.733 μg/mL), promoting apoptosis and tumor growth inhibition in mice, highlighting controlled release and tumor-targeting benefits [118]. Paclitaxel-functionalized gold nanoparticles exhibited higher anticancer activity against HT-29 and SiHa cells than free paclitaxel, emphasizing the role of nanocarrier-mediated drug delivery in enhancing cytotoxicity [119]. Nanostructured lipid carriers loaded with sclareol (size = 127 ± 16 nm and ζ = −27 ± 2 mV) outperformed solid lipid nanoparticles (size = 115 ± 5 nm and ζ = −22 ± 7mV) in cytotoxicity against MDA-MB-231 and HCT-116 cells (IC_50_ = 42 ± 3 and 75 ± 8 μM, respectively), demonstrating improved encapsulation efficiency and cellular delivery [120].

Carnosic acid-loaded PLGA nanoparticles induced higher ROS-mediated apoptosis in MDA-MB-231 cells compared to the free compound [121]. Combinatorial therapy of phytol and α-bisabolol in PLGA nanoparticles (size = 268 ± 54 nm) enhanced efficacy against A549 cells (IC_50_ = 14 μg/mL) [122]. Retinoic acid-loaded lipid–PLGA nanoparticles with CD133 aptamers (size = 129.9 nm and ζ = −18 mV) exhibited anticancer activity against H446 (IC_50_ = 5.5 ± 2.1 μg/mL) and A549 (IC_50_ = 8.5 ± 3.5 μg/mL) cells, demonstrating targeted delivery [123]. Chitosan-coated PLGA andrographolide nanoparticles induced G_1_ cell cycle arrest, enhanced cytotoxicity in MCF-7 cells, extended lifespan in Ehrlich ascites carcinoma-bearing mice by 78.08%, and reduced tumor mass by 68.21% [124]. Similarly, cetuximab-conjugated andrographolide-loaded chitosan–pectin nanocomposites (5 mg/kg) showed anticancer effects against 1,2-dimethylhydrazine-induced colon cancer of Swiss albino mice [125] and oridonin-loaded anisamide–lipid calcium phosphate nanoparticles (129.5 ± 23.7 nm, ζ potential = 23.6 ± 3.4 mV) demonstrated quicker release in acidic environments and prolonged release in vivo, both of which are advantageous for drug release in lung cancer [126]. In summary, these diterpenoid-based nanoformulations enhance anticancer efficacy through mechanisms including ROS-mediated apoptosis, cell cycle arrest, targeted drug delivery, and inhibition of tumor cell growth in vitro as well as tumor progression in vivo models.

### 6.4. Triterpenoids

Triterpenoids possess strong anticancer activity; however, their low aqueous solubility limits systemic availability and therapeutic efficacy. These limitations can be overcome using nanocarrier-based delivery systems such as liposomes, colloids, micelles, and nanoparticles [127]. Luteolin-incorporated electrospun polycaprolactone/gelatin nanocomposites exhibited cytotoxicity against ACHN (IC_50_ = 52.57 μg/mL) and HSC-3 (IC_50_ = 66.10 μg/mL) cells, demonstrating enhanced cellular uptake and controlled drug release. Self-assembled human serum albumin nanoparticles containing oleanolic acid and doxorubicin (size = ~140 nm) showed increased cellular association and synergistic chemotherapeutic activity in B16F10 and FaDuHTB-43 cells [128]. Oleanolic acid-loaded mPEG–PLGA nanoparticles (size = 200–250 nm) induced apoptosis in A549 and HepG2 cells, highlighting improved intracellular delivery and cytotoxicity [129]. PLGA nanoparticles co-loaded with oleanolic and ursolic acids (size = 217.98 ± 2.74 nm, ζ = 26.85 ± 0.49 mV) demonstrated significant cytotoxicity against Y-79 cells (IC_50_ = 16.6 ± 1.3 μM) [130], whereas magnetic nanoparticles of oleanolic acid (size = 40 nm) enhanced uptake and cytotoxic effects in A549 cells [131].

Ursolic acid-loaded methoxy poly(ethylene glycol)–polycaprolactone nanoparticles (size = 144.0 ± 4.0 nm, ζ = −0.99 ± 0.3) exhibited cytotoxicity in SGC7901 cells (IC_50_ = 46.0 ± 2.8 μM) via cell death induction, COX-2 inhibition, and caspase-3 activation [132]. Similarly, ursolic acid and ursolic acid-polymer micelles strongly suppressed HepG2 cell proliferation (IC_50_ = 43.2 ± 5.01 and 37.28 ± 2.44 μM, respectively) through enhanced apoptosis [133]. Niosomal ginsenoside Rh2 nanoparticles (size = 93.5 ± 2.1 nm) improved cytotoxicity and cellular uptake in PC3 cells (IC_50_ = 31.24 ± 0.75 μg/mL) [134]. Another study utilizing solid lipid nanoparticles (size = 73 nm) loaded with ganoderic acid demonstrated notable cytotoxicity against HepG2 cell lines during a 72-h incubation period, with an IC_50_ value of 25.10 µg/mL [135]. Similar results were shown in the significant cytotoxicity exhibited by ganoderic acid and ganoderic acid-loaded nano-lipidic carriers (GA-NLCs) (size = 156 nm) over a 48-h incubation period, with IC_50_ values of 40.0 and 32.61 µg/mL against HepG2 cells, respectively. Additionally, GA-NLCs may improve their anticancer activity in vivo by balancing biochemicals, antioxidants, and indicators of liver damage [136]. Functionalized graphene oxide carriers loaded with ganoderenic acid D reduced subcutaneous tumor mass by 27.27 ± 1.23% via activation of the intrinsic mitochondrial apoptotic pathway [137]. Overall, triterpenoid-based nanoformulations enhance anticancer potential by improving cellular uptake, inducing apoptosis, and exerting strong cytotoxicity across various cancer types, including liver, lung, stomach, prostate, and melanoma.

### 6.5. Tetraterpenoids

Numerous nanoencapsulation strategies have been developed to preserve the inherent properties of tetraterpenoids and enhance their anticancer potential [138]. Encapsulation of carotenoids in nanocarriers improves solubility, cell membrane penetration, intracellular uptake, bioaccessibility, and stability, thereby enhancing their cancer-preventive and therapeutic effects [139]. In prior studies, lycopene was incorporated into silver, iron, and gold nanoparticles with sizes ranging from 50 to 100 nm and HeLa, HT29, and COLO320DM cells were exposed to all nanoparticles at a concentration of 100 ug/mL . Gold nanoparticles (AuNPs) demonstrated 40.9% inhibition against HeLa and 41.47% inhibition against HT29, but silver nanoparticles (AgNPs) inhibited COLO320DM by 41.41% in the 3-(4,5-dimethylthiazol-2-yl)-2,5-diphenyltetrazolium bromide assay (MTT) assay. AgNPs demonstrated 82.68% inhibition for COLO320DM while AuNPs demonstrated 91.21% inhibition for HT29 and 87.98% inhibition for HeLa in the SRB study, indicating enhanced cytotoxicity through nanoparticle-mediated delivery [140]. Biosynthesized and functionalized AuNPs with hydroxylated 2-ketodeinoxanthin modulated gene expression and induced apoptosis, increasing anticancer efficacy [141]. 

Lutein-loaded PLGA–PEG–folate nanoparticles (188.0 ± 4.06 nm) enhanced lutein uptake by ~1.6–2-fold compared to PLGA nanoparticles or free lutein in SK-N-BE(2) cells, improving intracellular delivery [142]. Similarly, capsanthin-loaded diosgenin polyethylene glycol succinate 1000 (size = 508 ± 3) micelles demonstrated a noteworthy cytotoxic effect on MDA-MB-231 breast cancer cells, with an IC_50_ value of 3.10 ± 1.09 μg/mL by inducing apoptosis [143]. Another group of researchers found that chitosan–glycolipid nanocarriers of fucoxanthin (10 µM) demonstrated apoptotic action against Caco-2 cells [144]. Furthermore, another study showed that PLGA microspheres infused with fucoxanthin at a 1482.50 µg/mL concentration caused H1299 lung cancer cells to shrink, exhibit blebbing of the cell membranes, and develop apoptotic bodies [145]. Polypyrrole nanoparticles loaded with astaxanthin and coupled with bovine serum albumin (10–50 μg/mL) generated ROS in MDA-MB-231 cells under 808 nm laser irradiation, demonstrating potential for photothermal and photodynamic therapy as well as photoacoustic imaging [146]. These studies indicate that the tetraterpenoid-based nanoformulations enhance anticancer efficacy by promoting apoptosis, improving cellular uptake, and increasing cancer cell inhibition, with additional promise for theranostic applications.

**Table 3 cancers-17-03013-t003:** In vitro anticancer properties and characteristics of nanomaterials loaded with different terpenoids.

Type of Terpenoid	Compound	Type of Nanocarriers	Targeting Mediator/ Ligand	(Size, Shape and ζ Potential)	Cancer Type and Cells	IC_50_	Anticancer Effects	Mechanisms	Ref.
Monoterpenoids	Geraniol	Nanostructured lipid nanocarriers	None	110 nm, ζ potential = −10 mV, shape = NR	Lung cancer (A549)	1.5 mM	↓Cell viability	↑MMP loss	[147]
		Hyaluronic acid-based polymeric nanoconjugate	Hyaluronic acid	110 nm, Spherical, ζ potential = NR	Liver cancer (HepG2 and Huh7)	80 and 100 μM	↓Cell growth; ↑apoptosis; ↓cell proliferation	↑Bax; caspase-3; ↑caspase-9; ↓Bcl-2, ↓PARP	[148]
		Zinc–tin oxide/dextran/geraniol nanocomposites	None	197.40 nm, Agglomerated, ζ potential = NR	Colon cancer (HCT-116)	10 μg/mL	↑Cytotoxicity; ↑apoptosis	↑Caspase-3, ↑caspase-8; ↑caspase-9	[149]
	α-Pinene	Nanoemlusion	None	190 ± 8 nm, ζ potential = −10.4 ± 8 mV, shape = NR	Melanoma (A-375) and breast cancer (MCF-7)	106.19 and 168.02 μg/mL	↑Cytotoxicity; ↑apoptosis	↑Bax/Bcl-2 ratio	[150]
		Chitosan nanoparticles	None	102 ± 6 nm, ζ potential = 41.7 ± 1 mV, shape = NR	Melanoma (A-375)	76.4 µg/mL	↑Cytotoxicity	↑Bax/Bcl-2 ratio	[151]
	Linalool	Gold nanoparticles	CALNN peptide	5–20 nm, Spherical, ζ potential = NR	Breast cancer (MCF-7)	10 μg/mL	↑Cell growth; ↑apoptosis; ↓cell proliferation	↑ROS; ↓MMP; ↑caspase-8; ↑p53; ↓NF-κB	[152]
		Solid lipid nanoparticles	None	90–130 nm, Spherical, ζ potential = −4.0 mV	Liver (HepG2) and lung cancer (A549)	2 mM	↑Cytotoxicity	Not reported	[153]
		Gold nanoparticles	Glutathione + CALNN peptide	13 nm, shape and ζ potential = NR	Ovarian cancer (SKOV-3)	10 µg/mL	↑Cytotoxicity; ↑apoptosis; ↓cell proliferation	↑ROS; ↓MMP; ↑caspase-8; ↑p53; ↓NF-κB	[154]
	Menthol	Iron oxide nanoparticles	None	20–60 nm, shape and ζ potential = NR	Gastric cancer (AGS)	252 μg/mL	↑Cytotoxicity; ↑apoptosis	↑Caspase-8	[155]
		Chitosan gum	None	160 ± 15 nm, ζ potential = 43 ± 4 mV, shape = NR	Melanoma (A-375)	29 µg/mL	↑Cytotoxicity	Not reported	[156]
	Limonene	Chitosan nanoparticles	None	209 ± 13 nm, shape and ζ potential = NR	Melanoma (A-375) and breast cancer (MDA-MB-468)	30.24 and 650.70 µg/mL	↓Cell viability	Not reported	[157]
Sesquiterpenoids	β-caryophyllene	Silver nanoparticles	None	3.2 nm, Agglomerated, ζ potential = NR	Liver cancer (HepG2)	51.71 µg/mL	↓Cell growth	Not reported	[158]
		Silver nanoparticles	None	29.42 nm, Spherical, ζ potential = NR	Lung cancer (A549)	9.39 ± 0.08 g/mL	↓Cell proliferation	Not reported	[159]
	Artemisinin	Chitosan magnetic nanoparticles	None	349–445 nm, Spherical, ζ potential = −9.34 to −33.3 mV	Breast cancer (MCF-7)	25.61 ± 13 g/mL	↑Cytotoxicity; ↑apoptosis	Not reported	[160]
		Lipid nanoparticles	None	70 ± 20 nm, shape and ζ potential = NR	Triple negative breast cancer (MDA-MB-231)	7 ± 2 μm	↑Cytotoxicity	↓HER2; ↓survivin; ↓cyclin D1; ↓EGFR	[161]
		Mesoporous silica nanoparticles-loaded PLGA nanofibers	None	150–200 nm, shape and ζ potential = NR	Breast cancer (SK-BR-3)	55 μM	↑Cytotoxicity; ↑apoptosis	↑Bax; ↑caspase-3; ↑p53; ↓Bcl-2; ↓hTERT	[162]
	Farnesol	Chitosan-encapsulated nickel oxide, tin dioxide nanoparticles	None	34.8 nm, Agglomerated hexagonal structure, ζ potential = NR	Breast cancer (MDA-MB-231)	17.58 μg/mL	↑Cytotoxicity; ↑apoptosis; ↓cell proliferation	↓MMP; ↑ROS; ↑G2/M checkpoint arrest	[163]
Diterpenoids	Andrographolide	Solid lipid nanoparticles	None	Not reported	Head and neck squamous cell carcinoma (HN6 and HN30)	6.087 and 11.74 μg/mL	↑Cytotoxicity; ↑apoptosis; ↓cell proliferation	Not reported	[164]
		Albumin nanoparticles	None	100–200 nm, shape and ζ potential = NR	Cervical cancer (HeLa)	39.46 μg/mL	↑Cytotoxicity	Not reported	[165]
	Sclareol	Hyaluronan-coated PLGA nanoparticles	None	100–150 nm, ζ potential = −30 mV, shape = NR	Breast (MCF-7, MDA-MB468) and colon cancer (CaCo-2)	50 μM	↑Cytotoxicity	Not reported	[166]
		Solid lipid nanoparticles	None	88 nm, shape and ζ potential = NR	Lung cancer (A549)	19 μg/mL	↓Cell viability; ↑apoptosis	Not reported	[167]
	Carnosic acid	Albumin nanoparticles	None	97.29–144.26 nm, ζ potential = −21.03 mV, shape = NR	Colon (Caco-2) and breast cancer (MCF-7)	2.60 and 6.02 μg/mL	↓Cell viability; ↑apoptosis	↑G2/M checkpoint arrest; ↓COX-2; ↓Bcl-2; ↑GCLC; ↑p53	[168]
	Retinoic acid	Chitosan nanoparticles	None	313.23 ± 1.75 nm, ζ potential = 2.42 ± 0.04 mV, shape = NR	Breast cancer (MCF-7)	2.28 ± 0.02 µg/mL	↑Cytotoxicity; ↑apoptosis	↓Bcl-2; ↑caspase-3; ↑Bax; ↑cleaved PARP; ↑8-oxo-dG	[169]
		Glutenin Nanoparticles	Folic acid	~185 nm, Spherical, ζ potential = −3 mV	Breast cancer (MCF-7)	55.93 µg/mL	↑Cytotoxicity; ↑apoptosis; ↓cell proliferation	↓MMP; ↑ROS	[170]
		Solid lipid nanoparticles	None	140–150 nm, Spherical shape, ζ potential = −13 and −19 mV	Prostate cancer (LNCap)	200 μg/mL	↑Cytotoxicity; ↑apoptosis	Not reported	[171]
Triterpenoids	Squalene	Cisplatin-nanoprecipitated particles	None	128–160 nm, Spherical, ζ potential = NR	Colon cancer (HT-29 and KM-12)	8 μmol/L	↑Cytotoxicity; ↑apoptosis; ↓cell proliferation;	Not reported	[172]
		PLGA nanoparticles	None	Not reported	Colon cancer (Caco-2)	140 µg/mL	↑Cytotoxicity; ↑apoptosis	↑ROS generation	[173]
	Lupeol	Chitosan nanoparticles with cellulose acetate membrane	None	12 nm, shape and ζ potential = NR	Skin cancer (A431)	42.2 μg/mL	↑Cytotoxicity	Not reported	[174]
		Galactosylated liposomes	None	100 nm, shape and ζ potential = NR	Liver cancer (HepG2)	30 µM	↑Cytotoxicity; ↑apoptosis	↓p-Akt308; ↓p-Akt473 levels	[175]
	Oleanolic acid	Albumin nanoparticles	Cetuximab	171 ± 4.8 nm to 180 ± 3.7 nm, ζ potential = − 33.3 ± 3.4 mV, shape = NR	Lung cancer (A549)	4.34 ± 1.90 μg/mL	↑Cytotoxicity; ↑apoptosis; ↓cell proliferation	↑ROS generation; G_0_/G_1_ checkpoint arrest	[176]
	Ursolic acid	Gold PLGA nanoparticles	None	80 nm, Spherical, ζ potential = NR	Cervical cancer (SiHa, CaSki, and HeLa cells)	100 μM	↓Cell viability; ↑apoptosis; ↓cell proliferation; ↓cell migration ↓cell invasion	↑Bax; ↓Bcl-2; ↑caspase-3, ↑caspase-8; ↑capase-9; ↓procaspase-3; ↓procaspase-8; ↓procaspase-9; ↑p53; ↑fas; ↓cIAP	[177]
		Chitosan-coated PLGA nanoparticles	None	250 nm, Spherical, ζ potential = NR	Breast cancer (MCF-7 and MDA-MB-231)	26.74 and 40.67 μM	↑Cytotoxicity	Not reported	[178]
		PLGA-PEG nanoparticles	None	133.6 ± 0.7 nm, ζ potential = −22.6 ± 2.8 mV, shape = NR	Pancreatic ductal adenocarcinoma (AsPC-1 and BxPC-3)	11.7 ± 0.6 and 14.1 ± 2.2 μM	↑Cytotoxicity	Not reported	[179]
	Oridonin	Solid lipid nanoparticles	None	108.53 ± 10.92 nm, ζ potential = −37.97 ± 3.78 mV, shape = NR	Breast (MCF-7), lung (A549) and liver cancer (HepG2)	22.6, 25.3 and 30.1 μM	↓Cell viability; ↑apoptosis; ↓cell proliferation	↑G_2_/M checkpoint arrest; ↓G_1_/G_0_ checkpoint arrest;	[180]
	Ginsenoside	Liposomes combined with paclitaxel	None	77.71 ± 3.22 nm, ζ potential = −39.21 ± 1.03 mV, shape = NR	Gastric cancer (BGC-823)	0.04 μg/mL	↑Cytotoxicity; ↑apoptosis	Not reported	[181]
Tetraterpenoids	Lycopene	Eudragit RL 100 polymeric nanoparticles	None	62.10 ± 3.7 nm, Spherical	Prostate cancer (LNCaP and PC-3)	25.43 and 10.03 μg/mL	↑Cytotoxicity	Not reported	[182]
		Gold nanoemulsion	None	25.0 ± 4.2 nm, ζ potential = −32.2 ± 1.8 mV	Colon cancer (HT-29)	0.1 µM	↑Cytotoxicity; ↑apoptosis; ↓cell migration	↑Bax; ↓Bcl-2; ↓procaspase-8; ↓procaspase-3; ↓procaspase-9; ↑E-cadherin; ↓Akt; ↓NF-κB; ↓MMP-2; ↓MMP-9	[183]
	Carotenoid	Nanoemulsion	None	15.1 nm, shape and ζ potential = NR	Colon cancer (HT-29)	4.9 μg/mL	↑Cytotoxicity; ↑apoptosis; ↓cell proliferation	↑G_2_/M checkpoint arrest; ↑p53; ↑p21; ↓CDK1; ↓CDK2; ↓cyclin A; ↓cyclin B	[184]
	Lutein	Chitosan/alginate iron oxide nanoparticles	None	264 ± 6 nm, ζ potential = −13.3 ± 1.6 mV, shape = NR	Breast cancer (MCF-7)	4.12 ± 0.4 μg/mL	↓Cell viability	Not reported	[185]
	Fucoxanthin	Polyvinylpyrrolidone nanoparticles	None	<50 nm, shape and ζ potential = NR	Colon cancer (Caco-2)	20 μM	↓Cell viability; ↓cell migration	↑Pro-oxidative effect	[186]
	Astaxanthin	Chitosan nanoparticles	None	<400 nm, shape and ζ potential = NR	Melanoma (B16F10)	20 µg/mL	↑Antioxidant activity; ↓cell proliferation; ↓cell migration	Not reported	[187]
		Gold nanoparticles	None	60–120 nm, Spherical, ζ potential = NR	Breast cancer (MDA-MB-231)	50 μg/mL	↑Cytotoxicity; ↑apoptosis; ↓cell proliferation	Not reported	[188]
	β-Carotene	Solid lipid nanoparticles	None	203 ± 7.23 nm, ζ potential = −7.21 ± 0.82 mV, shape = NR	Breast cancer (MCF-7)	40 μg/mL	↑Antioxidant activity; ↑cytotoxicity	Not reported	[189]
		Solid lipid nanoparticles	None	111.78 nm, Spherical, ζ potential = −26.3 ± 1.3 mV, shape = NR	Breast cancer (MCF-7)	14.89 ± 0.02 μg/mL	↑Cytotoxic effect	Not reported	[190]

Symbols and abbreviations: ↑, increase/upregulation;  ↓, decrease/downregulation; Bax, Bcl-2-associated X protein; Bcl-2, B-cell lymphoma 2; CALNN, cysteine–alanine–leucine–asparagine–asparagine peptide; cIAP, cellular inhibitor of apoptosis protein; CDK, cyclin-dependent kinase; COX-2, cyclooxygenase-2; EGFR, epidermal growth factor receptor; GCLC, glutamate–cysteine ligase catalytic subunit; HER2, human epidermal growth factor receptor 2; hTERT, human telomerase reverse transcriptase; IC_50_, half maximal inhibitory concentration; MMP, mitochondrial membrane potential; MMP loss, loss of mitochondrial membrane potential; NF-κB, nuclear factor-κB; NR, not reported; p-Akt308/p-Akt473, phosphorylated Akt at threonine 308/serine 473; PARP, poly (ADP-ribose) polymerase; PLGA, poly (lactic-co-glycolic acid); ROS, reactive oxygen species; ζ potential, Zeta potential.

**Table 4 cancers-17-03013-t004:** In vivo anticancer properties and characteristics of nanomaterials loaded with different terpenoids.

Type of Terpenoid	Compound	Type of Nanocarriers	Targeting Mediator/ Ligand	Size, Shape and ζ Potential	Cancer Type (Model)	Dosage and Route of Administration	Antitumor Effects	Mechanisms	Ref.
Monoterpenoids	Geraniol	Hyaluronic acid-based polymeric nanoconjugate	Hyaluronic acid	110 nm, ζ potential = −10 mV, shape = NR	Liver cancer (H22 cells-injected mice)	Intravenous injection, 1.0 mg geraniol/kg	↓Tumor size; ↓tumor weight; ↓tumor volume	↓Ki-67	[148]
		Zinc–tin oxide/dextran/geraniol nanocomposites	None	197.40 nm, Agglomerated, ζ potential = NR	Colon cancer (DMH-induced rats)	20 and 40 mg/kg	↓Tumor volume; ↓tumor size; ↓tumor incidence	↓COX-2	[149]
Diterpenoids	Andrographolide	PLGA nanocapsulation	None	163 nm, Nanospheres, ζ potential = −57.85 mV	Cervical cancer (HeLa cells-injected mice)	10mg/kg	↓Tumor size	Not reported	[191]
	Carnosic acid	Liposomes	Transferrin	97.06 ± 3.389 nm, ζ potential = 2.55 ± 1.26 mV, shape = NR	Liver cancer (HepG2- and SMMC-7721-transplanted BALB/c nude mice)	Intraperitoneal injection, 2 mg/kg	↓Tumor growth; ↓tumor volume	↑Caspase-3; ↑caspase-9; ↑Bax; ↑Bad; ↑PARP; ↓Bcl-2	[192]
	Oridonin	PEG-PLGA nanoparticles	None	100 nm, ζ potential = −5 mV, shape = NR	Breast cancer (MCF-7 cells-injected BALB/c nude mice)	Intraperitoneal injection, 10 mg/kg	↓Tumor size; ↓tumor volume; ↓angiogenesis	NR	[193]
Triterpenoids	Ursolic acid	Chitosan nanoparticles	Folate	160nm, ζ potential = 39.3mV, shape = NR	Breast cancer (MCF-7 cells-injected BALB/c mice)	Intraperitoneal injection, 12.5mg/kg	↓Tumor size; ↓tumor weight	NR	[194]
		Gold PLGA nanoparticles	None	80 nm, Spherical, ζ potential = NR	Cervical cancer (SiHa and HeLa cells-injected nude mice)	Intraperitoneal injection, 20 mg/kg	↓Tumor size; ↓tumor weight	NR	[177]
Tetraterpenoids	Lycopene	N-isopropylacrylamide with N-vinyl2-pyrrolidone poly(ethyleneglycol)monoacrylate copolymeric nanoparticles	None	<100 nm, Spherical, ζ potential = NR	Melanoma (TPA-induced Swiss albino mice)	Topical treatment, 1 μg/mL	NR	↓COX-2, ↓Bcl-2, ↑Bax	[195]
	Astaxanthin	Ethylcellulose nanoparticles	None	185 nm, Spherical, ζ potential = NR	Oral cancer (DMBA- induced Syrian hamsters)	Oral treatment, 15 mg/kg	↓Tumor size; ↓tumor growth	Cyclin D1; ↓Bcl-2, ↑Bax; ↑caspase-3; ↑caspase-9	[196]
	β-Carotene	Thiolated chitosan -lithocholic acid nanomicelles	None	<300 nm, ζ potential = +27.0 mV, shape = NR	Skin cancer (DMBA- induced BALB/c mice)	Topical treatment, 1 mg/mL	↓Tumor size; ↓tumor weight	NR	[197]

Symbols and abbreviations:↑, increase/upregulation;  ↓, decrease/downregulation; Bad, Bcl-2-associated death promoter; Bax, Bcl-2 associated X protein; Bcl-2, B-cell lymphoma 2; COX-2, cyclooxygenase 2; DMBA, 7,12-dimethylbenz[a]anthracene; DMH, 1,2-dimethylhydrazine; NR, not reported; PARP, poly (ADP-ribose) polymerase; PEG-PLGA, polyethylene glycol–poly(lactic-co-glycolic acid); PLGA, poly(lactic-co-glycolic acid); TPA, 12-O-tetradecanoylphorbol-13-acetate; ζ potential, Zeta potential.

## 7. Challenges and Emerging Trends in Nanoformulated Terpenoids

Although these innovative nanodrug delivery methods of terpenoids have many benefits, there are also several drawbacks in terms of real-world applications. Several of these challenges include the specific description of these nanoproducts, their toxicity and safety characterizations, and the absence of efficient control. Two significant obstacles are the increased cost and manufacturing complexity, as the synthesis and characterization of nanoformulations require several expensive tools. Furthermore, although there are many approved nanomedicines on the list, their clinical uses are limited because of the lack of a specific regulatory standard for the formulation and assessment of these nanoproducts. Since nanotechnology has diverse applications, more comprehensive and updated regulatory standards are required. The governing bodies of multiple countries must work together to develop precise and rigorous rules that can handle major safety issues and guarantee the creation of safe and advantageous nanomedicines for humankind [198].

Nanoformulated terpenoid compounds encapsulated within nanoparticles have shown promise in enhancing cancer diagnosis and therapy. However, several challenges and limitations hinder their widespread clinical applications. One major concern is biocompatibility and toxicity, as variations in nanoparticle size, shape, and surface chemistry can result in unexpected interactions with biological systems, potentially causing side effects on healthy tissues [199]. Additionally, stability and aggregation pose substantial hurdles, particularly for polymeric nanoparticles commonly used to encapsulate terpenoids. Under physiological conditions, these nanoparticles are susceptible to aggregation, potentially undermining colloidal stability, reducing their bioavailability, and hindering their effectiveness in drug delivery and imaging applications [200].

Another limitation is the lack of precise targeting of cancer cells. Many nanoformulated terpenoids exhibit non-specific distribution throughout the body, which not only reduces the therapeutic levels at the tumor site but simultaneously elevates the risk of off-target effects on healthy tissues. This underscores the urgent need for advanced and specific targeting strategies, such as ligand-mediated or stimuli-responsive systems, to increase the selectivity and diagnostic accuracy of these formulations [200]. Furthermore, reliance on the EPR effect for tumor targeting presents another challenge. Although the EPR effect facilitates passive accumulation of nanoparticles in tumor tissues, its effectiveness is highly inconsistent due to tumor heterogeneity, vascular abnormalities, and variations in the tumor microenvironment. These factors can significantly limit the predictability and uniformity of nanoparticle accumulation across different tumor types and patient populations, leading to variable therapeutic and diagnostic outcomes [201].

Nanosized drug delivery methods generate concerns about short- and long-term toxicities and biocompatibility, which should be evaluated by specialized in vivo toxicology investigations in suitable animal models [202]. The important factors are the cost of these medicines and the distribution of nanomedicines globally at the lowest levels of care (government hospitals and primary healthcare). Furthermore, additional investigation is crucial to accurately characterize and interpret these toxicities and their effect on the environment, human body, and human physiology. Researchers are seriously concerned about the ethical application of nanomedicine in society as well as the safety concerns [203]. Addressing these challenges requires ongoing research to develop nanoformulations with improved stability, biocompatibility, and targeted delivery capabilities to increase the efficacy of terpenoid-based cancer diagnostics.

New developments in nanoformulations lead to the formation of multipurpose nanosystems that can perform imaging, diagnostics, and therapy concurrently, paving the way for personalized medicine strategies. Furthermore, the combination of stimulus-responsive nanoparticles and smart materials offers the possibility of regulated drug release, which can be triggered by external forces or physiological factors. Their translational ability is further enhanced by the creation of biocompatible and biodegradable nanocarriers, which will reduce toxicity and increase drug absorption [204]. Moreover, bioinformatics techniques are essential for applications including drug targeting, analysis of protein–protein interactions, and identification of compound–target networks and cancer treatment pathways. A growing trend in bioinformatics, information technology, and omics is the development of resources that provide information on herbal formulations and the bioactive chemicals of medicinal plants. Numerous initiatives do exist as databases, such as the Collective Molecular Activities of Useful Plants (CMAUP), SymMap, Indian Medicinal Plants, Phytochemistry and Therapeutics (IMPPAT), and the Indian Medicinal Plants Database (IMPLAD). Additionally, experts have created new approaches for analyzing the pharmacokinetic characteristics of medications and phytochemicals using in silico methods. These methods can also be applied to in silico screening, elucidation of plausible mechanisms of action, and drug discovery of phytoconstituents [205].

Drugs made using nanocarriers may present a number of difficulties, one of which is obtaining regulatory approval to use clinical trials. As a result, artificial intelligence combined with these nanocarriers may challenge the existing framework for these kinds of innovative interventional treatments. In contrast to wet-lab methods, artificial intelligence (AI) tools offer a logical framework for screening a large range of chemicals using models, which reduces the need for time and resources. Along with other aspects, not limited to drug-target identification, bioactivity, de novo drug design, physicochemical property estimation, and quantitative structure–activity relationship (QSAR)-based drug development, AI has also been used to identify drug toxicity, drug loading, and surface-level characterization. AI-based formation of nanoformulated medications and their targets may enhance the platform for creating innovative anticancer treatments [206]. AI has been used to predict treatment efficacy and to design, characterize, and manufacture drug delivery nanosystems [207]. The integration of in silico modeling or AI and machine learning tools may successfully address the challenges related to terpenoid-loaded nanoformulations in cancer therapy. This enables precise design and optimization of targeted drug delivery systems, thereby improving the therapeutic efficiency and specificity of anticancer compounds (Figure 8).

## 8. Conclusions and Future Directions

Terpenoids have been recognized as crucial components of current and future potential anticancer drugs. The use of nanotechnology in cancer treatments has demonstrated promise in addressing the issues with conventional treatment strategies, such as those related to instability, high toxicity, poor water solubility, low absorption, multidrug resistance, low bioavailability, and low specificity. Additionally, using multi-compartmental nanocarriers allows the concurrent delivery of several medications, which makes it easier to co-administer herbal and other medications for synergistic effects. In our review, the different classes of terpenoids and their anticancer mechanisms were elaborately discussed. Furthermore, terpene-based nanoformulations are known to be an efficient tool for delivering drugs, as they improve the solubility, bioavailability, biocompatibility, and help to overcome the drawbacks of plant-based terpenoids used for cancer treatment. It is an extremely difficult task for researchers and manufacturers to create terpenoid-based nanoformulations and optimize their dosage either alone or in mixture with other anticancer drugs. Nonetheless, researchers are certain that terpene-based nanoformulations will soon find a place in the arsenal of cancer treatment and tumor therapies. In the future, comprehensive investigations are needed to evaluate the synthesis, drug delivery mechanisms, potential toxic effects, and safety aspects of terpenoid-based nanoformulations through rigorous in vitro and in vivo studies.

## Figures and Tables

**Figure 1 cancers-17-03013-f001:**
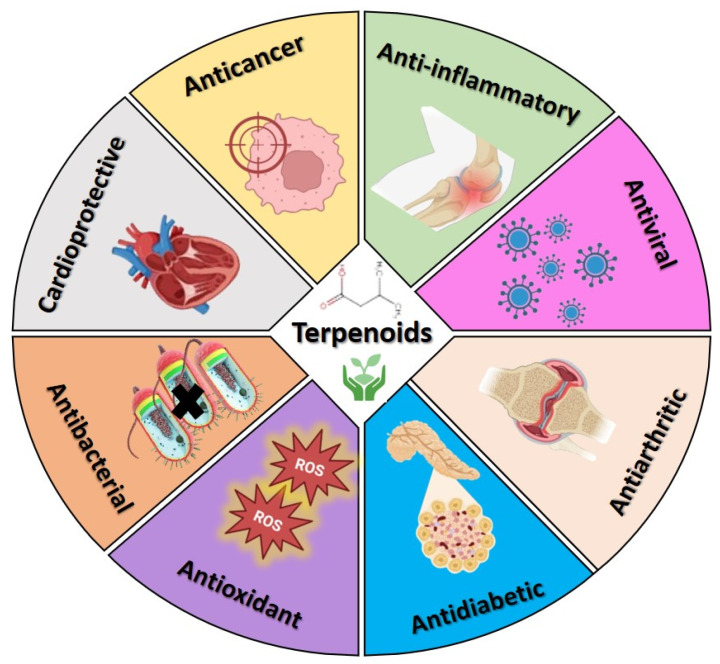
Schematic illustration of various biological applications of terpenoids.

**Figure 2 cancers-17-03013-f002:**
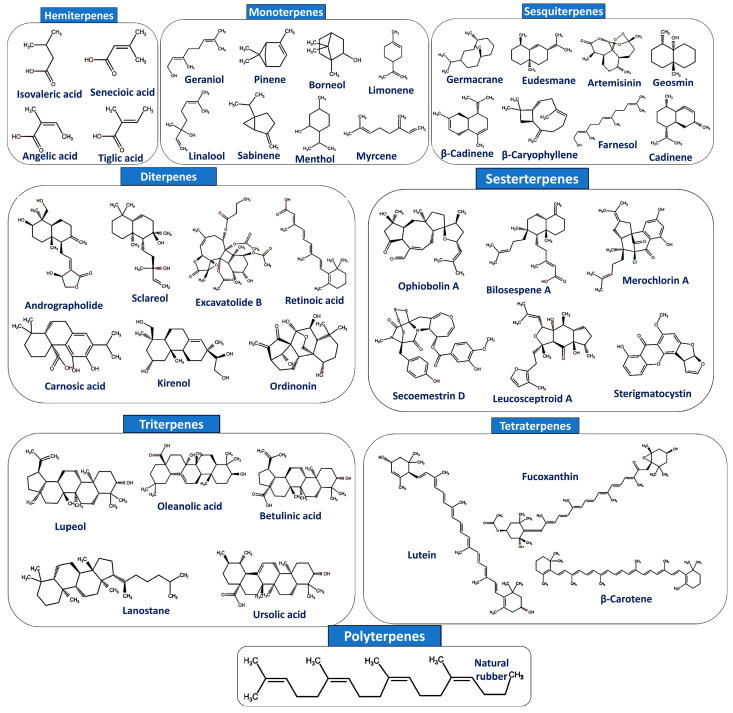
The chemical structures of different classes of terpenoids. Chemical structures were drawn using ChemAxon’s MarvinSketch software (version 17.21.0).

**Figure 3 cancers-17-03013-f003:**
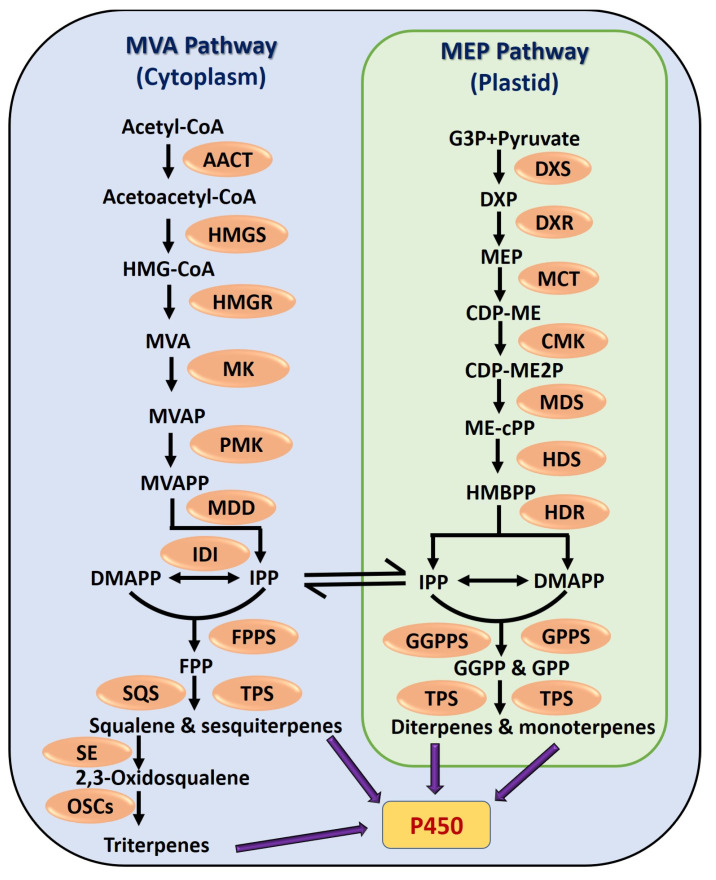
Terpenoid biosynthesis via the cytosolic MVA and plastidic MEP pathways, both producing isoprenoid precursors as building blocks for diverse terpenoids. Abbreviations: DMAPP, dimethylallyl diphosphate; DXP, 1-deoxy-D-xylulose 5-phosphate; DXR, 1-deoxy-D-xylulose 5-phosphate reductoisomerase; FPP, farnesyl diphosphate; FPPS, farnesyl diphosphate synthase; GGPP, geranylgeranyl diphosphate; GGPPS, geranylgeranyl diphosphate synthase; GPP, geranyl diphosphate; GPPS, geranyl diphosphate synthase; HMGR, 3-hydroxy-3-methylglutaryl-CoA reductase; HMGS, 3-hydroxy-3-methylglutaryl-CoA synthase; IDI, isopentenyl diphosphate isomerase; IPP, isopentenyl diphosphate; MEP, methylerythritol phosphate; MVA, mevalonate; MK, mevalonate kinase; MDD, mevalonate diphosphate decarboxylase; OSCs, oxidosqualene cyclases; PMK, phosphomevalonate kinase; SE, squalene epoxidase; TPS, terpene synthase.

**Figure 4 cancers-17-03013-f004:**
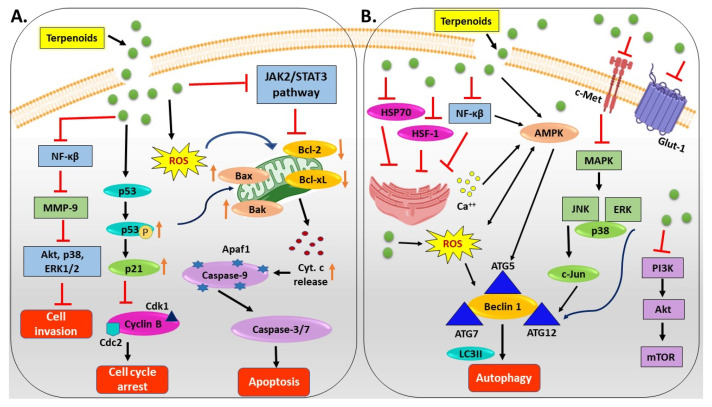
Anticancer mechanism of terpenoids. (**A**) Terpenoids trigger cell cycle arrest and apoptosis via p53, Bax/Bak, and caspases. (**B**) Terpenoids promote autophagy via AMPK/ROS and inhibit PI3K/Akt/mTOR. Abbreviations: AMPK, AMP-activated protein kinase; Apaf1, apoptotic protease activating factor 1; ATG, autophagy-related protein; Bax, Bcl-2-associated X protein; Bak, Bcl-2 homologous antagonist/killer; Bcl-2, B-cell lymphoma 2; Bcl-xL, B-cell lymphoma-extra-large; c-Jun, cellular Jun proto-oncogene; c-Met, mesenchymal–epithelial transition factor; Cdc2, cell division cycle protein 2; Cdk1, cyclin-dependent kinase 1; Cyt. c, cytochrome c; ERK, extracellular signal-regulated kinase; Glut-1, glucose transporter 1; HSF-1, heat shock factor 1; HSP70, heat shock protein 70; JAK2/STAT3, Janus kinase 2/signal transducer and activator of transcription 3; JNK, c-Jun N-terminal kinase; LC3II, microtubule-associated protein 1A/1B-light chain 3; MAPK, mitogen-activated protein kinase; MMP-9, matrix metalloproteinase-9; mTOR, mammalian target of rapamycin; NF-κβ, nuclear factor-κβ; PI3K, phosphatidylinositol-3-kinase; ROS, reactive oxygen species.

**Figure 5 cancers-17-03013-f005:**
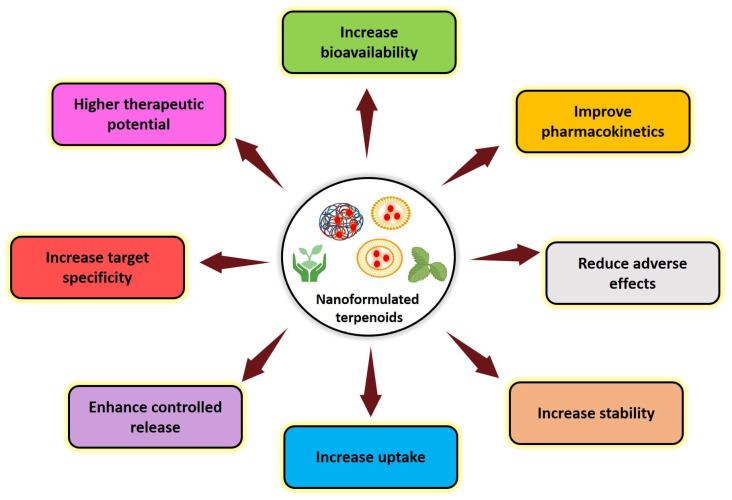
Overview of the various advantages of nanoformulated terpenoid-based anticancer drugs.

**Figure 6 cancers-17-03013-f006:**
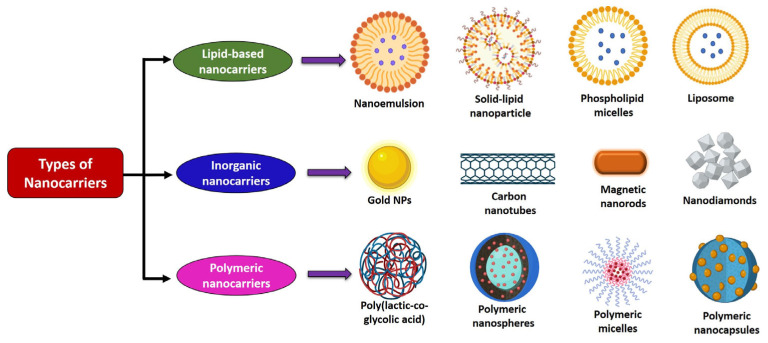
Schematic representation of different nanocarriers used in targeted anticancer drug delivery.

**Figure 7 cancers-17-03013-f007:**
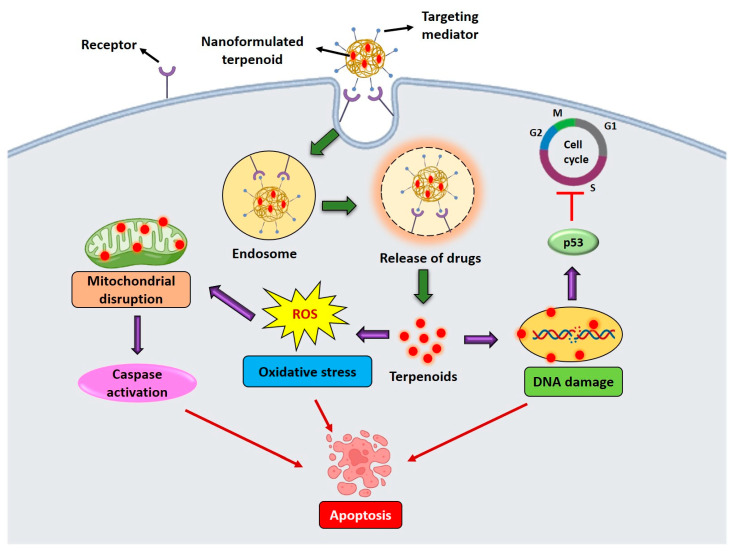
Schematic diagram showing the mechanisms of action of nanoformulated terpenoids.

**Figure 8 cancers-17-03013-f008:**
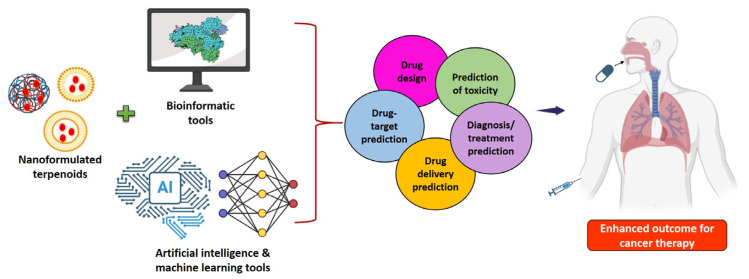
Schematic diagram of a synergistic approach combining nanoformulated terpenoids with advanced bioinformatics tools and artificial intelligence to optimize cancer therapy.

**Table 2 cancers-17-03013-t002:** Comparative overview of nanocarriers for terpenoid delivery in cancer therapy.

Type of Nanocarrier	Route of Administration	Advantages	Disadvantages	Ref.
Self-emulsifying drug delivery systems	Oral	Enhance solubility and absorption of hydrophobic terpenoids	Limited drug loading and requires high surfactant content	[89]
Polymeric nanoparticles	Oral and IV	Sustained/controlled release; biodegradable; and targeted delivery	Complex synthesis; morphology and size dependency; and burst release possible	[90]
Pickering emulsions	Oral	Surfactant-free; good physical stability	Scale-up challenges	[91]
Liposomes	Oral, IV, and topical	Encapsulate both hydrophilic and lipophilic terpenoids; biocompatible; enhance solubility and systemic availability	Stability issues; drug leakage; and high cost	[87]
Solid lipid nanoparticles/nanostructured lipid carriers	Oral and IV	Encapsulate hydrophobic terpenoids; high stability; controlled release; and reduced toxicity	Limited drug loading and risk of drug expulsion	[86]
Ethosomes, phytosomes, niosomes, and invasomes	Oral, transdermal, and topical	Enhance skin permeability, absorption, and stability	Limited systemic application and variable reproducibility	[86,87]
Inorganic nanoparticles	IV	Easy surface modification and theranostic potential	Long-term toxicity concerns and poor biodegradability	[83]

Abbreviations: IV, intravenous.

## Data Availability

Not applicable.

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
