# Peer review of "Nanoformulated Terpenoids in Cancer: A Review of Therapeutic Applications, Mechanisms, and Challenges"

_cancers, 2025, doi:10.3390/cancers17183013_

Round 1
Reviewer 1 Report
Comments and Suggestions for Authors
Dear Editor,
This review is well-written and provides valuable information on the nanoformulation of terpenoids as promising anticancer agents for targeted therapy. The manuscript is both interesting and well-structured, with a clear presentation of the topic, the figures are also well-presented and clear so it can be accepted in present form.
Kind regards
Author Response
The authors of this manuscript express their sincere thanks to the Assistant Editor and reviewers for the critical assessment of this work. The authors have acted upon the Assistant Editor’s and reviewers' recommendations, which have significantly enhanced the quality of this manuscript. All modifications incorporated in the manuscript are highlighted in red font. A “point-by-point” response to each comment is outlined below. We have also indicated the exact location of our manuscript, subject to modifications by citing page and line numbers.
General comments:
This review is well-written and provides valuable information on the nanoformulation of terpenoids as promising anticancer agents for targeted therapy. The manuscript is both interesting and well-structured, with a clear presentation of the topic, the figures are also well-presented and clear so it can be accepted in present form.
Response:
We sincerely thank the reviewer for their positive and encouraging comments regarding the quality, clarity, and presentation of our manuscript. We truly appreciate the reviewer’s recommendation for acceptance in the present form.
Reviewer 2 Report
Comments and Suggestions for Authors
Authors have presented a review emphasizing “Nanoformulation of terpenoids as promising anticancer agents for targeted therapy”. Overall, the paper is well written however there are a few places where the unnecessary data is incorporated. However, these don't detract from the meaning of the text but a careful proofread could address these issues and improve the flow of the text. In my opinion, the manuscript must be improved by incorporating these suggestions:
- Authors have to improve the abstract section by leading their story with terpenoids as well as
- Similar case for the introduction section. Authors are suggested to improve the introduction section by providing a brief content on cancer epidemiology and therapies presented from line number 60-89 and incorporate more content related to natural compounds nanoformulations from recent articles like; https://doi.org/10.2147/IJN.S520580, https://doi.org/10.1007/s00210-025-03937-y, https://doi.org/10.3390/molecules26061764. Similarly provide specific content of terpenoids nanoformulations from line number 115-144. Also add and explain the novelty of this review in introduction section.
- In my opinion there is no requirement of explaining terpenoids classification, biosynthesis section. There are already wide range of data already exist related to this; authors can provide latest reference for these. Authors must delete these segment or brief in one paragraph to avoid unnecessary segment and this will also distract from the main theme i.e. nanoformulations of natural phytocompounds like terpenoids.
- Also, there is no need of explaining the different types of nanoformulations, authors should compress the segment 6.
- Authors are suggested to provide mechanistic diagram for explaining the anticancer properties of terpenoids based nanoformulations.
- Authors must restructured the manuscript and thoroughly proofread the manuscript to remove unnecessary data included in this manuscript. This will help readers to better understand the anticancer potential of terpenoids nanoformulations.
- Authors are suggested to carefully check the grammatical errors and spelling mistakes throughout the manuscript.
The English could be improved to more clearly express the research.
Author Response
The authors of this manuscript express their sincere thanks to the Assistant Editor and reviewers for the critical assessment of this work. The authors have acted upon the Assistant Editor’s and reviewers' recommendations, which have significantly enhanced the quality of this manuscript. All modifications incorporated in the manuscript are highlighted in red font. A “point-by-point” response to each comment is outlined below. We have also indicated the exact location of our manuscript, subject to modifications by citing page and line numbers.
General comments:
Authors have presented a review emphasizing “Nanoformulation of terpenoids as promising anticancer agents for targeted therapy”. Overall, the paper is well written however there are a few places where the unnecessary data is incorporated. However, these don't detract from the meaning of the text but a careful proofread could address these issues and improve the flow of the text. In my opinion, the manuscript must be improved by incorporating these suggestions:
Response:
We thank the reviewer for the constructive feedback and positive remarks on our manuscript. We carefully reviewed the text and revised unnecessary data to enhance clarity and improve the overall flow of the manuscript.
Specific comments:
Comment 1:
Authors have to improve the abstract section by leading their story with terpenoids as well as Similar case for the introduction section. Authors are suggested to improve the introduction section by providing a brief content on cancer epidemiology and therapies presented from line number 60-89 and incorporate more content related to natural compounds nanoformulations from recent articles like; https://doi.org/10.2147/IJN.S520580, https://doi.org/10.1007/s00210-025-03937-y, https://doi.org/10.3390/molecules26061764. Similarly provide specific content of terpenoids nanoformulations from line number 115-144. Also add and explain the novelty of this review in introduction section.
Response:
Thank you for the insightful suggestion. The abstract was modified as per the suggestion (page 1, line 33 to page 2, line 55). We added an overview of cancer epidemiology and therapies (page 3, lines 65-70). We have incorporated recent advances on natural compound nanoformulations from the cited articles and added specific details on terpenoid-based nanoformulations (page 4, lines 124-130; pages 4, line 140 to page 5, line 143; and page 5, lines 156-164). The novelty of the review was included in the introduction section (page 5, lines 174-185).
Comment 2:
In my opinion there is no requirement of explaining terpenoids classification, biosynthesis section. There are already wide range of data already exist related to this; authors can provide latest reference for these. Authors must delete these segment or brief in one paragraph to avoid unnecessary segment and this will also distract from the main theme i.e. nanoformulations of natural phytocompounds like terpenoids. Also, there is no need of explaining the different types of nanoformulations, authors should compress the segment 6.
Response:
Thank you for this constructive feedback. We reduced the terpenoid classification and biosynthesis into a brief paragraph (page 5, line 186 to page 6, line 214). We have now compressed Section 5 by providing only a concise overview of nanoformulation types with relevant references (page 11, line 344 to page 12, line 386).
Comment 3:
Authors are suggested to provide mechanistic diagram for explaining the anticancer properties of terpenoids based nanoformulations.
Response:
Thank you for the valuable comment. A mechanistic diagram (Figure 7) illustrating the anticancer properties of nanoformulated terpenoids has been added (page 15).
Comment 4:
Authors must restructured the manuscript and thoroughly proofread the manuscript to remove unnecessary data included in this manuscript. This will help readers to better understand the anticancer potential of terpenoids nanoformulations.
Response:
We appreciate your time in reviewing our manuscript and observing the flow between sections. We have restructured the manuscript and thoroughly proofread it to remove unnecessary data and improve clarity. Specifically, the earlier Section 2 (Classification of terpenoids) and Section 3 (Biosynthesis of terpenoids) have been merged into a single section for better flow. In addition, Section 5 was emphasized to focus on processes of preparing terpenoid nanoformulations.
Comment 5:
Authors are suggested to carefully check the grammatical errors and spelling mistakes throughout the manuscript.
Response:
We appreciate this comment. We have carefully reviewed the manuscript and corrected all grammatical errors and spelling mistakes to improve readability and accuracy. All edits and changes are highlighted in red color.
Reviewer 3 Report
Comments and Suggestions for Authors
Dear Editor,
The manuscript presents a comprehensive and valuable contribution to the interdisciplinary fields of oncology, natural product chemistry, and nanotechnology. It effectively integrates these disciplines to substantiate the potential of nanoformulated terpenoids in cancer therapy. The manuscript’s strengths significantly outweigh its minor weaknesses. With minor revisions to enhance the discussion on translational challenges and the application of artificial intelligence, this paper will undoubtedly become a significant and impactful publication.
- I recommend revising your article title to highlight the key findings of your study on this nanoformulation. Compared to the current title, the present version is too vague and unpredictable. From the title alone, I cannot form a hypothesis about the content, nor can I tell that this is a review article without reading the abstract.
- In the abstract, please include a brief description of the methodology you used to develop and structure this review.
- The abstract currently only outlines what will be discussed in the review. There are no actual results presented. I suggest including at least one highlighted study preferably the most compelling example of nanomodification of terpenoids with potent anticancer effects as part of the results section in the abstract.
- In the introduction, between the 4th and 5th paragraphs, please insert a section describing the limitations of terpenoids that can be addressed by nanoformulation. This will improve the logical flow of your introduction.
- In the introduction (lines 137–140), please include a statement about targeted drug delivery.
- Regarding Table 2: Since your study aims to discuss nanoparticle targeted drug delivery systems, it would be valuable to include a dedicated column for the types of targeting mediators/ligands that enable targeted delivery in each nanoformulation. I noticed some are mentioned in the nanoparticle types column, but this is not ideal. Presenting them in a separate column would be more academically sound, as they are the core components enabling targeted delivery.
- Sections 7.1 to 7.5 currently have very limited discussion and read more like a presentation of results mostly comparing study outcomes without explaining why certain studies show higher potency while others do not. There is little exploration of the mechanisms underlying how these nanoformulations enhance the cytotoxic effects of terpenoids. I suggest adding:
- A dedicated discussion section that provides rational explanations for the observed phenomena, including all factors influencing the results, presented in a detailed and academic manner.
- A separate “Mechanism Insight” section specifically describing how nanotargeted terpenoid formulations act on cancer cells. In your current draft, the mechanism discussion appears to be for pure terpenoids rather than the nanoformulated forms.
- Optional suggestion: In a supplementary section, before the main chapters, include a concise overview of nanoparticle drug delivery systems and nanoparticle targeted drug delivery systems to set the context for your review.
Thank you
Comments on the Quality of English LanguageTo ensure the highest professional standard upon submission, it is advisable to undergo a comprehensive proofread, meticulously examining minor grammatical errors and typographical inaccuracies..
Author Response
The authors of this manuscript express their sincere thanks to the Assistant Editor and reviewers for the critical assessment of this work. The authors have acted upon the Assistant Editor’s and reviewers' recommendations, which have significantly enhanced the quality of this manuscript. All modifications incorporated in the manuscript are highlighted in red font. A “point-by-point” response to each comment is outlined below. We have also indicated the exact location of our manuscript, subject to modifications by citing page and line numbers.
General comments:
The manuscript presents a comprehensive and valuable contribution to the interdisciplinary fields of oncology, natural product chemistry, and nanotechnology. It effectively integrates these disciplines to substantiate the potential of nanoformulated terpenoids in cancer therapy. The manuscript’s strengths significantly outweigh its minor weaknesses. With minor revisions to enhance the discussion on translational challenges and the application of artificial intelligence, this paper will undoubtedly become a significant and impactful publication.
Response:
Thank you for the positive and encouraging feedback. We appreciate your recognition of the manuscript’s strengths and have revised the discussion to further elaborate on translational challenges and the potential applications of artificial intelligence in advancing terpenoid-based nanoformulations for cancer therapy.
Specific comments:
Comment 1:
I recommend revising your article title to highlight the key findings of your study on this nanoformulation. Compared to the current title, the present version is too vague and unpredictable. From the title alone, I cannot form a hypothesis about the content, nor can I tell that this is a review article without reading the abstract.
Response:
We thank the reviewer for this constructive comment. The title has been revised (New title: Nanoformulated Terpenoids in Cancer: A Review of Therapeutic Applications, Mechanisms, and Challenges) to be more specific and informative, clearly reflecting the focus and actual content of a review article.
Comment 2:
In the abstract, please include a brief description of the methodology you used to develop and structure this review.
Response:
We are grateful for this observation. Since this manuscript is a narrative review and not a systematic one, we have revised the abstract to provide a brief overview of the main sections and key messages of the review.
Comment 3:
The abstract currently only outlines what will be discussed in the review. There are no actual results presented. I suggest including at least one highlighted study preferably the most compelling example of nanomodification of terpenoids with potent anticancer effects, as part of the results section in the abstract.
Response:
We acknowledge this suggestion. Since this is a review article, the abstract has been framed to present a concise collective evaluation of multiple relevant studies rather than emphasizing a single study, in order to maintain balance and reflect the overall scope of the work.
Comment 4:
In the introduction, between the 4th and 5th paragraphs, please insert a section describing the limitations of terpenoids that can be addressed by nanoformulation. This will improve the logical flow of your introduction.
Response:
We have considered this thoughtful recommendation. The limitations of terpenoids and how nanoformulation can address them was added as per the suggestion (page 4, lines 124-130).
Comment 5:
In the introduction (lines 137–140), please include a statement about targeted drug delivery.
Response:
Thank you for this insightful comment on the introductory section. A statement highlighting the role of nanoformulated terpenoids in targeted drug delivery has now been added (page 5, lines 176-181).
Comment 6:
Regarding Table 2: Since your study aims to discuss nanoparticle targeted drug delivery systems, it would be valuable to include a dedicated column for the types of targeting mediators/ligands that enable targeted delivery in each nanoformulation. I noticed some are mentioned in the nanoparticle types column, but this is not ideal. Presenting them in a separate column would be more academically sound, as they are the core components enabling targeted delivery.
Response:
We appreciate this insightful suggestion. We have revised Tables 2 and 3 by adding a separate column for targeting mediators/ligands to highlight their role in targeted delivery. This change enhances clarity and provides a more structured presentation for better reader understanding on significance of targeting components in nanoformulations.
Comment 7:
Sections 7.1 to 7.5 currently have very limited discussion and read more like a presentation of results mostly comparing study outcomes without explaining why certain studies show higher potency while others do not. There is little exploration of the mechanisms underlying how these nanoformulations enhance the cytotoxic effects of terpenoids. I suggest adding:
- A dedicated discussion section that provides rational explanations for the observed phenomena, including all factors influencing the results, presented in a detailed and academic manner.
- A separate “Mechanism Insight” section specifically describing how nanotargeted terpenoid formulations act on cancer cells. In your current draft, the mechanism discussion appears to be for pure terpenoids rather than the nanoformulated forms.
Response:
We agree with the comment and the comparison of study outcomes was included in all the section 6.1 to 6.5 and we added a discussion in last of each paragraph explaining how nanoformulations enhance terpenoid anticancer effects (page 16, lines 462-465; page 16, lines 499-502; page 17, lines 534-537; page 18, lines 571-573; and page 19, lines 603-605). A dedicated “Mechanism Insight” section has now been added to specifically describe how nanoformulated terpenoid formulations act on cancer cells (page 14, line 387 and page 15, line 427).
Comment 8:
Optional suggestion: In a supplementary section, before the main chapters, include a concise overview of nanoparticle drug delivery systems and nanoparticle targeted drug delivery systems to set the context for your review.
Response:
We acknowledge this suggestion. A concise overview of nanoparticle drug delivery systems and targeted drug delivery systems has already been incorporated in the Introduction (page 4, line 140 to page 5, line 164), where we discuss their role in improving solubility, stability, bioavailability, targeted delivery, and therapeutic efficacy.
Reviewer 4 Report
Comments and Suggestions for Authors
Comments
- The captions of Figures 2–6 are excessively long, particularly those of Figures 3, 4, and 6. It is recommended to shorten the captions and transfer the detailed explanations into the main text.
- Line 267: Please remove the phrase ‘are bioactive substances mainly derived from plants,’ as this information has already been mentioned in lines 90–94.
- In Section 4, please provide more details regarding the mechanisms by which terpenoids interact with cancers, for example, their interactions with cancer receptor sites. In addition, the explanation currently included in the legend of Figure 4 should be revised and incorporated into the main text under this section.
- In Section 6, the authors stated that there are three main groups of nanocarriers. The subheadings 6.1, 6.2, and 6.3 should be revised to ‘Lipid-based nanocarriers,’ ‘Inorganic nanocarriers,’ and ‘Polymeric nanocarriers,’ respectively. Furthermore, each subsection should provide more detailed explanations regarding the types commonly employed for terpenoid delivery and a discussion of their respective advantages and disadvantages to terpenoids.
Author Response
The authors of this manuscript express their sincere thanks to the Assistant Editor and reviewers for the critical assessment of this work. The authors have acted upon the Assistant Editor’s and reviewers' recommendations, which have significantly enhanced the quality of this manuscript. All modifications incorporated in the manuscript are highlighted in red font. A “point-by-point” response to each comment is outlined below. We have also indicated the exact location of our manuscript, subject to modifications by citing page and line numbers.
Comment 1:
The captions of Figures 2–6 are excessively long, particularly those of Figures 3, 4, and 6. It is recommended to shorten the captions and transfer the detailed explanations into the main text.
Response:
We appreciate the reviewer’s insightful observation. The captions of Figures 2–6 have been shortened; however, inclusion of a list of abbreviations (as per standard style) may make the legends appear longer.
Comment 2:
Line 267: Please remove the phrase ‘are bioactive substances mainly derived from plants,’ as this information has already been mentioned in lines 90–94.
Response:
Thank you for the thoughtful suggestion. We have removed the phrase "are bioactive substances mainly derived from plants" (page 10, line 286).
Comment 3:
In Section 4, please provide more details regarding the mechanisms by which terpenoids interact with cancers, for example, their interactions with cancer receptor sites. In addition, the explanation currently included in the legend of Figure 4 should be revised and incorporated into the main text under this section.
Response:
We appreciate the reviewer’s insightful feedback. Section 4 has been revised to include detailed mechanisms, and the content from the Figure 4 legend has been integrated into the main text (page 8, line 242 to page 9, line 256).
Comment 4:
In Section 6, the authors stated that there are three main groups of nanocarriers. The subheadings 6.1, 6.2, and 6.3 should be revised to ‘Lipid-based nanocarriers,’ ‘Inorganic nanocarriers,’ and ‘Polymeric nanocarriers,’ respectively. Furthermore, each subsection should provide more detailed explanations regarding the types commonly employed for terpenoid delivery and a discussion of their respective advantages and disadvantages to terpenoids.
Response:
We appreciate the reviewer’s comment. In line with feedback from another reviewer, Section 5 was concisely revised, and a comparative table (Table 2) has been added along with their respective advantages and disadvantages for terpenoid delivery, effectively addressing both recommendations (page 12, lines 356-380).
Reviewer 5 Report
Comments and Suggestions for Authors
- 1.The Introduction section mentions various plant-derived anticancer compounds such as phenols, alkaloids, and terpenoids, but it does not analyze the unique advantages of terpenoid compounds, failing to highlight why terpenoids are chosen as the core component of nanoformulations. It is recommended to add comparisons to emphasize the advantages of terpenoid compounds in nanoformulations.
- The section "4. Mechanistic insights into the anticancer properties of terpenoids" summarizes the overall anticancer mechanisms of terpenoid compounds, but it does not analyze the mechanistic differences among different categories of terpenoids. It is recommended to add comparisons of different terpenoids to enhance the practicality of the article.
- 4.In the section "Mechanistic Insights into the Anticancer Properties of Terpenoids," the mechanisms of cell cycle, apoptosis, and autophagy are individually discussed, but the synergistic effects of these mechanisms in exerting anticancer actions are not explained, leading to a "fragmented" understanding of the mechanisms. It is suggested to enhance the systematic understanding of the anticancer effects of terpenoids.
- 5. Challenges with natural terpenoids and advantage of nanoformulations: The text generally mentions that nanoformulations address the three major limitations of terpenoids but does not specify the differences in nano strategies, resulting in "insufficient targeting" of the solutions. It is recommended to include specific examples of solutions to enhance the credibility of the article.
- 5.Challenges with natural terpenoids and advantages of nanoformulations: This section only emphasizes the advantages of nanoformulations but fails to objectively mention potential risks and limitations, lacking the author's perspective on the limitations of terpenoid nanoformulations.
- 6.Various Processes of Preparing Nanoformulations Containing Terpenoids: It is recommended to add a subsection that compares the advantages, disadvantages, and suitable types of terpenoids for different carriers in a tabular format. Additionally, the authors' opinions and recommendations on carrier selection should be provided in conjunction with specific cancer types to strengthen the connection between nanocarriers and terpenoids in cancer treatment.
7. In Figure 2, the structures of the compounds are not clear enough, and the key functional groups are not labeled. In Table 1, the "Biological activities" column is labeled as "Attracts insects for pollination, and precursors of plant hormones," but it is explicitly mentioned earlier that tetraterpenes (such as carotenoids) possess "immunomodulatory and anticancer activities," which is logically inconsistent. In Table 2, the column (Size, Shape & ζ potential) does not include all required elements. For example, Farnesol only mentions particle size and morphology, without indicating zeta potential, which does not align with the column header "(Size, Shape & ζ potential)" that requires "particle size, morphology, and zeta potential." There is missing data information, and it is recommended to check the data integrity of this table.
Author Response
The authors of this manuscript express their sincere thanks to the Assistant Editor and reviewers for the critical assessment of this work. The authors have acted upon the Assistant Editor’s and reviewers' recommendations, which have significantly enhanced the quality of this manuscript. All modifications incorporated in the manuscript are highlighted in red font. A “point-by-point” response to each comment is outlined below. We have also indicated the exact location of our manuscript, subject to modifications by citing page and line numbers.
Comment 1:
The Introduction section mentions various plant-derived anticancer compounds such as phenols, alkaloids, and terpenoids, but it does not analyze the unique advantages of terpenoid compounds, failing to highlight why terpenoids are chosen as the core component of nanoformulations. It is recommended to add comparisons to emphasize the advantages of terpenoid compounds in nanoformulations.
Response:
Thank you for this valuable comment. We have incorporated a comparative discussion highlighting the unique advantages of terpenoids over other phytochemicals (page 4, lines 130-137).
Comment 2:
The section "4. Mechanistic insights into the anticancer properties of terpenoids" summarizes the overall anticancer mechanisms of terpenoid compounds, but it does not analyze the mechanistic differences among different categories of terpenoids. It is recommended to add comparisons of different terpenoids to enhance the practicality of the article.
Response:
We appreciate the valuable suggestion. While individual terpenoid categories and their anticancer mechanisms have been extensively studied in previous reports (Reference No: 1 Kamran S, Sinniah A, Abdulghani MA, Alshawsh MA. Therapeutic potential of certain terpenoids as anticancer agents: a scoping review. Cancers. 2022 Feb 22;14(5):1100, Reference No: 2 Yoon YE, Jung YJ, Lee SJ. The anticancer activities of natural terpenoids that inhibit both melanoma and non-melanoma skin cancers. International journal of molecular sciences. 2024 Apr 17;25(8):4423), our article primarily focuses on nanoformulated terpenoids. Accordingly, we have emphasized the anticancer effects of terpenoids in their nanoformulated forms, as per earlier reviewer recommendations, to provide a more practical and application-oriented perspective relevant to nanocarrier-based therapies (page 14, lines 387-417).
Comment 3:
In the section "Mechanistic Insights into the Anticancer Properties of Terpenoids," the mechanisms of cell cycle, apoptosis, and autophagy are individually discussed, but the synergistic effects of these mechanisms in exerting anticancer actions are not explained, leading to a "fragmented" understanding of the mechanisms. It is suggested to enhance the systematic understanding of the anticancer effects of terpenoids.
Response:
We appreciate this thoughtful observation. In the revised version, we have highlighted how cell cycle arrest, apoptosis, and autophagy act in a coordinated and synergistic manner through shared signaling pathways, providing a more systematic understanding of terpenoids’ anticancer effects (page 9, lines 257-270).
Comment 4:
Challenges with natural terpenoids and advantage of nanoformulations: The text generally mentions that nanoformulations address the three major limitations of terpenoids but does not specify the differences in nano strategies, resulting in "insufficient targeting" of the solutions. It is recommended to include specific examples of solutions to enhance the credibility of the article.
Response:
We thank the reviewer for this suggestion. We have added specific examples of different nanocarrier strategies with terpenoids to address this comment (page 10, lines 309-318).
Comment 5:
Challenges with natural terpenoids and advantages of nanoformulations: This section only emphasizes the advantages of nanoformulations but fails to objectively mention potential risks and limitations, lacking the author's perspective on the limitations of terpenoid nanoformulations.
Response:
We thank the reviewer for this valuable feedback. To address it, we have added a separate section titled “7. Challenges and emerging trends in terpenoid-loaded nanoformulations” (page 28, line 636 to page 29, line 689) where we have discussed potential risks and limitations, including toxicity, stability issues, non-specific targeting, cost, manufacturing complexity, and regulatory concerns.
Comment 6:
Various Processes of Preparing Nanoformulations Containing Terpenoids: It is recommended to add a subsection that compares the advantages, disadvantages, and suitable types of terpenoids for different carriers in a tabular format. Additionally, the authors' opinions and recommendations on carrier selection should be provided in conjunction with specific cancer types to strengthen the connection between nanocarriers and terpenoids in cancer treatment.
Response:
We appreciate the insightful suggestion. A comparative table highlighting advantages, disadvantages, and terpenoid suitability for different nanocarriers, along with our opinions and recommendations for cancer-specific applications, has been incorporated in the revised manuscript (pages 12, lines 369-380 and page 13, Table 2).
Comment 7:
In Figure 2, the structures of the compounds are not clear enough, and the key functional groups are not labeled. In Table 1, the "Biological activities" column is labeled as "Attracts insects for pollination, and precursors of plant hormones," but it is explicitly mentioned earlier that tetraterpenes (such as carotenoids) possess "immunomodulatory and anticancer activities," which is logically inconsistent. In Table 2, the column (Size, Shape & ζ potential) does not include all required elements. For example, Farnesol only mentions particle size and morphology, without indicating zeta potential, which does not align with the column header "(Size, Shape & ζ potential)" that requires "particle size, morphology, and zeta potential." There is missing data information, and it is recommended to check the data integrity of this table.
Response:
We thank the reviewer for this observation. Figure 2 (page 6) has been replaced with high-resolution images (DPI = 3000). The higher DPI significantly improves image clarity, enhances the visibility of fine structural details, and ensures sharper readability of labels and features, thereby making the figure more appropriate for accurate interpretation and publication quality.
We have re-checked Table 3 (page 20) and harmonized the “Size, Shape & ζ potential” column, reporting all three metrics when available, and clearly marking morphology or ζ-potential as “NR/NA” where studies did not assess or report them.
Round 2
Reviewer 4 Report
Comments and Suggestions for Authors
The revision is satisfactory for me.
Reviewer 5 Report
Comments and Suggestions for Authors
Thank you for inviting me to conduct the second-round review of this manuscript. After carefully reading the revised version and the author's responses, I have observed significant improvements in the research content, language expression, and presentation of research results, which have made the paper's structure much clearer and more coherent. I acknowledge the progress made in the manuscript.
Given the author's comprehensive responses and appropriate revisions, I recommend accepting this manuscript for publication.